# Activating p53 abolishes self-renewal of quiescent leukaemic stem cells in residual CML disease

Mary T. Scott [1,5], Wei Liu[1,5], Rebecca Mitchell[2], Cassie J. Clarke [2], Ross Kinstrie[1], Felix Warren [2], Hassan Almasoudi [1,3], Thomas Stevens[1], Karen Dunn [2], John Pritchard [1], Mark E. Drotar[1,2], Alison M. Michie [2], Heather G. Jørgensen [2], Brian Higgins[4], Mhairi Copland [2] & David Vetrie [1] ✉

Whilst it is recognised that targeting self-renewal is an effective way to functionally impair the quiescent leukaemic stem cells (LSC) that persist as residual disease in chronic myeloid leukaemia (CML), developing therapeutic strategies to achieve this have proved challenging. We demonstrate that the regulatory programmes of quiescent LSC in chronic phase CML are similar to that of embryonic stem cells, pointing to a role for wild type p53 in LSC self-renewal. In support of this, increasing p53 activity in primitive CML cells using an MDM2 inhibitor in combination with a tyrosine kinase inhibitor resulted in reduced CFC outputs and engraftment potential, followed by loss of multi-lineage priming potential and LSC exhaustion when combination treatment was discontinued. Our work provides evidence that targeting LSC self-renewal is exploitable in the clinic to irreversibly impair quiescent LSC function in CML residual disease – with the potential to enable more CML patients to discontinue therapy and remain in therapy-free remission.

Due to the success of tyrosine kinase inhibitor (TKI) therapy, chronic myeloid leukaemia (CML), a rare myeloproliferative disease arising through the acquisition of BCR::ABL1 in a haematopoietic stem cell (HSC)[1], is predicted to become the most prevalent form of leukaemia within 30 years[2]. Increased prevalence of CML will severely impact healthcare budgets[3] and more and more patients will be faced with the physical (i.e., drug side effects), social and psychological issues of living with a chronic form of leukaemia[4,5]. Current therapy-free remission (TFR) rates in CP CML, when TKI are safely stopped or de-escalated, are disappointingly low at ~10–15% of CP patients[6–8]. This is due to the inability of TKI to eradicate quiescent leukaemic stem cells (LSC) which persist in almost all chronic phase patients[9–11] as residual disease and which can drive disease recurrence when TKI therapy is stopped.

Developing novel strategies to improve TFR rates are an important area of clinical need. Of particular interest, are strategies focussed on selectively disabling self-renewal which provides a means to functionally inactivate LSC, ultimately leading to their exhaustion or inability to drive disease recurrence. Whilst pathways regulating self-renewal of LSC in CML have been examined over the last two decades[12], developing clinically-tractable strategies to target these pathways have proven challenging. Strikingly, LSC share a number of features with embryonic stem (ES) cells, including dependence on c-MYC and PRC1/2[13–19], low levels of wild type (WT) p53[18,20,21] and their proliferative potential[22–24]. Based on these similarities, we hypothesized that LSC co-opt the very primitive regulatory circuitry found in ES cells allowing them to self-renew and remain quiescent.

[1]Wolfson Wohl Cancer Research Centre, School of Cancer Sciences, University of Glasgow, Glasgow, UK. [2]Paul O'Gorman Leukaemia Research Centre, Institute of Cancer Sciences, University of Glasgow, Glasgow, UK. [3]Department of Clinical Laboratory Sciences, College of Applied Medical Sciences, Najran University, Najran, Saudi Arabia. [4]Genentech Inc, South San Francisco, CA, USA. [5]These authors contributed equally: Mary T. Scott, Wei Liu. ✉ e-mail: David.Vetrie@glasgow.ac.uk

Here we show that CML LSC have global regulatory circuitry which reflects that found in ES cells, particularly when the LSC are quiescent. Importantly, the ES cell circuitry of quiescent LSC points to a low WT p53 state. We demonstrate that upregulating p53 through MDM2 inhibition efficiently disables and exhausts quiescent LSC in the presence of TKI. This defines a role for WT p53 as a critical regulator of self-renewal in LSC, and provides a therapeutically tractable route to improving TFR rates in CP CML.

## Results

### LSC signatures reflect BCR::ABL1-dependent ES cell circuitry

To provide molecular evidence of the similarities between LSC and ES cells, we first examined published bulk transcriptome data from peripheral blood CD34$^+$CD38$^-$ LSC and HSC[19]. We found ES cell-like signatures previously shown to be enriched in cancer cells[13,14] were also enriched in LSC compared with HSC (Fig. 1a) whilst signatures of adult stem cells[25,26] were depleted in LSC (Fig. 1a; Supplementary Fig. 1a). Additionally, LSC also displayed enrichment of signatures for c-MYC, E2F, DREAM complex targets[27], and BRCA1/CHEK2 networks, the genes from which overlapped with those from ES cell signatures (Supplementary Fig. 1b). To explore the significance of these ES-related signatures, we examined the mean mRNA level of each signature across 19 clusters of peripheral blood derived CD34$^+$CD38$^-$ HSC or LSC obtained from single cell RNA-sequencing (scRNA-seq) (Supplementary Fig. 1c). A greater proportion of LSC compared to HSC showed high expression of ES-related signatures (64% in LSC versus 27% in HSC) (Fig. 1b) suggesting that they were a dominant feature of LSC molecular circuitry.

We examined the molecular architecture of ES-related circuitry in CML cells further using bulk and scRNA-seq datasets obtained from BCR::ABL1$^-$ and BCR::ABL1$^+$ primitive bone marrow (BM) cells from a BCR::ABL1 transgenic murine model of CML and from CML patients[28]. Pearson correlation coefficient analysis of genes found in the ES-related signatures identified clusters of genes whose expression was highly correlated in BCR::ABL1$^+$ cells (shown in yellow in the heatmap of Fig. 1c, d). BCR::ABL1$^-$ cells lacked these clusters and displayed lower mean Pearson correlation coefficients for the same gene set (histograms of Fig. 1c, d), demonstrating that BCR::ABL1 directs this ES-like molecular circuitry. We then projected publicly available transcription factor (TF) binding and histone modification patterns derived from murine ESC[17] onto our CML ES-related circuitry (which we termed ESC-REG) and performed correlation analysis to identify over-represented patterns of TF and histone modification co-occupancy at target genes. In both primitive BCR::ABL1$^+$ murine BM or CML BM cells, genes from the ESC-REG were over-represented for patterns associated with either Myc-or Nanog–regulatory complexes, both highly characteristic of ES cell circuitry[14,17] (Fig. 1e).

### Quiescent LSC are highly enriched for ES cell circuitry and have low *TP53*

We next explored how the CML ESC-REG circuitry was modulated in quiescent and cycling cells. Cell cycle predictions from scRNA-seq analysis of CML and normal CD34$^+$CD38$^-$ cells isolated from peripheral blood showed LSC had a noticeably higher proportion of cells in S or G/2 M when compared to HSC (Supplementary Fig. 2a, b) – consistent with cell cycle profiles observed in ESC[29]. Furthermore, using the same approach as described above (Fig. 1c, d), we identified highly correlated ESC-REG circuitry in cycling or quiescent CD34$^+$ cells isolated from peripheral blood of CML patients[23] (Fig. 2a panel i; Supplementary Fig. 2c; Supplementary Data 2). Notably, the circuitry within quiescent cells was significantly more characteristic of ES cells than that found in cycling cells (Fig. 2a, panel ii), and was predicted to be highly over-represented for Myc and Nanog complex co-occupancy at ESC-REG genes (Fig. 2b).

Next, from the same scRNA-seq dataset, we identified 500 exemplar LSC from each phase of the cell cycle, including those

showing low/absent expression of *MKI67*, compatible with the quiescent state. The *MKI67*$^{high}$ cycling LSC showed noticeably higher expression of 1101 ESC-REG genes than *MKI67*$^{low}$ quiescent LSC, and this included higher expression of a large set of canonical cell cycle regulators (Fig. 2c). Conversely, 101 ESC-REG genes were upregulated in the quiescent LSC, and these included 63 ribosomal protein genes, the upregulation of which is also characteristic of ES cells[30]. Given the overall expression profiles of the regulatory circuitry we identified, we defined quiescent and cycling LSC as having ESC-REG$^{low}$ and ESC-REG$^{high}$ molecular phenotypes respectively.

The gene encoding p53, *TP53*, is a member of the ESC-REG found in both BM derived and peripheral blood derived LSC (Supplementary Data 2). Quiescent ESC-REG$^{low}$ LSC showed lower expression of *TP53* and lower expression of the majority of the p53 target genes within the ESC-REG (Fig. 2d). Strikingly, TKI treatment of LSC in vitro for up to 7 days, downregulated ES cell signatures (Supplementary Fig. 2d), and reduced expression of the majority of 464 deregulated ESC-REG genes, including *TP53* and p53 targets (Fig. 2e; Supplementary Data 3), suggesting that TKI treatment preserves, or possibly enhances, the ESC-REG$^{low}$*TP53*$^{low}$ phenotype. Based on our in silico analysis described above, we hypothesized that quiescent LSC, like ESC, require low levels of WT p53 to preserve their capacity to self-renew and that activation of p53 would target quiescent LSC and abolish their self-renewal capacity.

### Mdm2 inhibition impairs LSC survival in vitro in a p53-dependent manner

To test our hypothesis, we used a potent MDM2 inhibitor (MDM2i), idasanutlin (IDASA) to upregulate p53 in CML cells in vitro and in vivo and examined how this affected CFC, engraftment, and multilineage potentialities, and maintenance of a CD34$^+$CD38$^-$ cell pool (all qualitative or quantitative metrics of self-renewing LSC). Using CML cell lines that were either p53 WT or null (BV173 or K562/KCL22 respectively), we confirmed that the reduction of cell counts and CFC outputs, as a consequence of IDASA treatment, was dependent on WT p53 function (Fig. 3a; Supplementary Fig. 3a). Western blot analysis of IDASA treated BV173 cells also resulted in upregulation of p53 when used alone or in combination with the TKI nilotinib (NIL) (Fig. 3b). Similarly, treatment of CML CD34$^+$ cells from patients in chronic phase or blast phase CML (all p53$^{WT}$) with IDASA and/or NIL was effective at selectively increasing apoptosis and reducing CFC and LT-CIC outputs particularly when used in combination (Fig. 3c, d; Fig. S3b).

To provide evidence for p53 activation at the molecular level, we performed bulk RNA-seq on CML CD34$^+$ cells treated with these drugs for 24 or 72 hrs (Supplementary Data 4) and, in parallel, we identified p53 targets in CML cells by ChIP-sequencing (Supplementary Fig. 3c; Supplementary Data 5). NIL treatment alone for 24 hrs led to limited increases in p53 signalling as measured by statistical enrichment of p53 molecular signatures and increased expression of p53 target genes (Fig. 3e). IDASA treatment alone for 24 hrs was able to drive more robust increases in p53 signalling. However, increased p53 signalling was sustained for up to 72 hrs only when cells were treated with both NIL and IDASA in combination (Fig. 3e, f). This provided an explanation why the combination of NIL plus IDASA was more effective in vitro than IDASA treatment alone.

As highlighted previously (Fig. 2e), NIL treatment alone was accompanied by repression of ESC-REG genes (24 hrs) followed by repression of p53 signalling (72 hrs) (Fig. 3e). Curiously, treatment with NIL plus IDASA targeted repression of ESC-REG genes beyond that achieved with NIL alone, despite upregulating p53 targets (Venn diagram, Fig. 3e). To reconcile these findings, we showed that p53 may exert negative feedback on the ESC-REG; combined NIL plus IDASA treatment of the WT p53 BV173 CML cell line led to increased p53 while decreasing c-MYC, a key regulator of the ESC-REG (Fig. 3b). Thus, combined NIL plus IDASA treatment synergized to reinforce the ESC-REG$^{low}$ state whilst simultaneously activating p53 – suggesting that

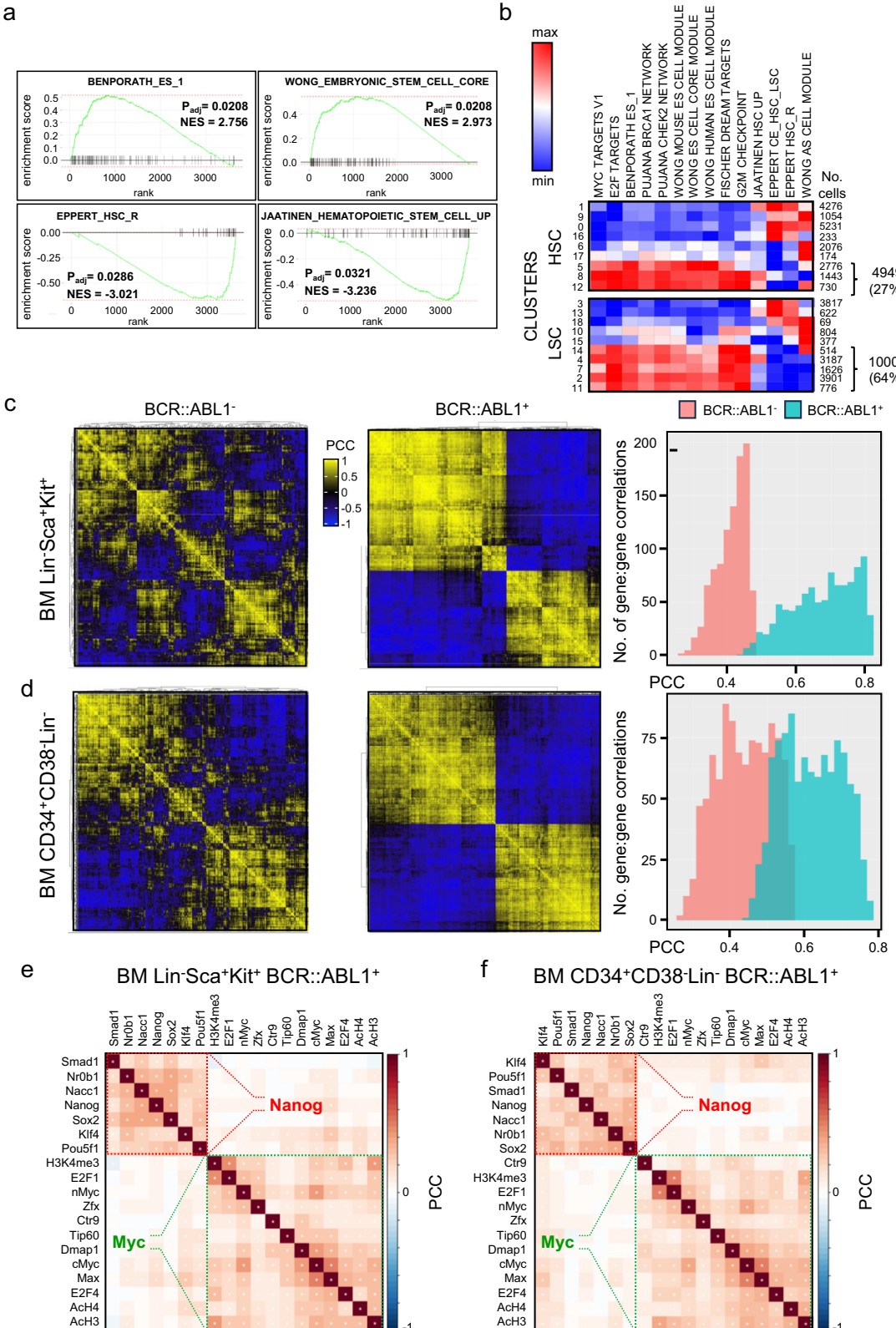

MDM2 inhibition can activate p53 irrespective of the repressive effect on the ESC-REG found in quiescent LSC.

## Loss of colony forming potential of leukaemic cells upon p53 activation in vivo

Using a patient-derived xenograft (PDX) model (Fig. 4a), we observed the functional consequences of p53 activation on chronic phase CML LSC engrafted in murine bone marrow (BM). Mice were treated with and without IDASA for 5 days at the beginning of a 28 day cycle of treatment with nilotinib (NIL) to mirror MDM2i drug scheduling used in a trial setting (NCT02545283). This approach allowed us to also examine whether increasing p53 levels on a TKI background would also impair LSC function. We analysed the effects of these agents on human cell engraftment at both 5 and 28 days in separate cohorts (cohorts 1

**Fig. 1 | LSC display a BCR::ABL1-dependent ES cell regulatory network (ESC-REG). a** ES cell and adult stem cell molecular signatures (MSigDB) identified by gene set enrichment analysis of LSC versus HSC (E-MTAB-2581[19]). $P_{adj}$ derived by Wilkinson method (EGSEA) with Benjamini-Hochberg multiple testing correction. **b** Heatmaps showing mean expression of ES cell, adult stem cell and other related molecular signatures (MSigDB) in HSC and LSC clusters (see also Supplementary Fig. 1). Numbers of cells in each cluster are shown at the right. Total numbers of HSC and LSC (and percentages) showing high expression of ES cell signatures are bracketed. **c** Heatmaps depicting Pearson correlation coefficients (PCC) resulting from gene-pair analysis of candidate ESC-REG genes in murine BM-derived Lin⁻Sca⁺Kit⁺ expressing BCR::ABL1, compared with those for the same genes in murine BM-derived Lin⁻Sca⁺Kit⁺ lacking BCR::ABL1 expression. The histogram (right) shows the distribution of PCC across the gene set in the two conditions. **d** PCC for gene-pair analysis of candidate ESC-REG genes in human BM-derived CD34⁺CD38⁻Lin⁻ BCR::ABL1 ⁺ cells compared with those for the same genes in CD34⁺CD38⁻Lin⁻ BCR::ABL1 ⁻ cells (all from the same chronic phase CML patients). The histogram (right) shows the distribution of PCC across the gene set in the two conditions. **e** Heatmap shows PCC analysis for predicted TF and epigenetic co-occupancy of ESC-REG genes in murine BM Lin⁻Sca⁺Kit⁺ BCR::ABL1⁺ cells. **f** Heatmap shows PCC analysis for predicted TF and epigenetic co-occupancy of ESC-REG genes in human BM CD34⁺CD38⁻Lin⁻ BCR::ABL1 ⁺ cells. Source data are provided as a Source Data file.

and 2 in Fig. 4a). In parallel, we performed CFC assays on human CD45⁺ cells isolated from the BM of these PDX mice and calculated outputs based on cell equivalents plated. We observed no significant effects on the levels of human CD45⁺CD34⁺CD38⁻ LSC across any of the treatment arms compared with vehicle (VEH) at either time point (Supplementary Fig. 4a; Fig. 4b). However, at the end of the 28 day treatment, CFC outputs from mice treated with IDASA alone or in combination with NIL, showed marked reductions when compared to those from the VEH and NIL arms (Fig. 4b ii), despite showing no differences in levels of CD45⁺CD34⁺CD38⁻ LSC at that timepoint (Fig. 4b i). These data demonstrated that at the end of treatment, the remaining leukaemic cells were functionally impaired and had lost their colony-forming potential as a consequence of p53 activation.

## p53 activation decreases the LSC through time during a treatment-free period

To monitor the longer-term effects of MDM2 inhibition/p53 upregulation on engrafted chronic phase CML LSC in a scenario where therapy is removed (mimicking TKI discontinuation), we performed serial analysis of human cells at the end of a 28-day cycle of NIL +/- IDASA treatment, and then again after a 28-day treatment-free period (cohort 3, Fig. 4a). No significant reductions in human LSC cell levels were detectable at the end of 28 days between any of the treatment arms (Fig. 4C; Supplementary Fig. 4b). However, by the end of the 28-day treatment-free period, levels of CD45⁺, CD45⁺CD34⁺ and CD45⁺CD34⁺CD38⁻ LSC showed noticeable reductions relative to VEH in the mice exposed to IDASA, or IDASA plus NIL, compared to those mice treated with NIL only (Fig. 4c; Supplementary Fig. 4c). Similar results were observed when we examined the effects of IDASA +/- NIL on LSC from blast phase CML in PDX mice (Fig. 4d; Supplementary Fig. 4d, e). Importantly, chronic and blast phase LSC levels at the end of the treatment-free period, were lower than those found at the end of treatment, particularly in mice treated with IDASA in combination with NIL when compared to those treated with NIL alone (Fig. 4e). These results are consistent with p53 activation promoting LSC loss/exhaustion - an effect observed 51 days after MDM2i treatment was withdrawn from the combination treatment arm (Supplementary Fig. 4e).

## p53 activation impairs engraftment potential of LSC

To examine whether MDM2i mediated p53 activation impairs LSC engraftment potential, we used a transplantable model of CML-like disease (SCL-tTA-BCR-ABL-inducible transgene) (Supplementary Fig. 5a)[31]. Treatment with either NIL or the combination of NIL plus RG7112, an MDM2i that targets murine MDM2[32], showed excellent control of CML-like disease in primary recipient mice (Supplementary Fig. 5b–d) similar to the results reported by others[33]. When primitive cells were examined in the bone marrow (BM) within the donor CD45.2 cell population, there was a significant reduction in Lin⁻Sca⁺Kit⁺ (LSK) populations in the combination treatment compared to NIL alone and in the RG7112 only arm compared to VEH (Supplementary Fig. 5e). Within the CD45.2 long-term (LT)-HSC population, however, combined NIL plus RG7112 did not lead to significant reductions below that

achieved by treatment with NIL alone (Supplementary Fig. 5f). To determine whether leukaemic LT-HSC engraftment potential had been impaired, we transplanted whole BM pools that had received either NIL or combination treatment into secondary cohorts; each recipient mouse was transplanted with 750 CD45.2 LT-HSC. Strikingly, we observed a significant decrease in the number of recipient mice that successfully engrafted CD45.2 cells from the combination arm compared to the NIL alone arm (2 of 7 versus 6 of 7; $p = 0.05$; Supplementary Fig. 5g). These data lend further support to the view that p53 activation impairs LSC function on a TKI background.

## p53 activation targets the most primitive LSC and correlates with increased ROS production

We extended our study to ask whether LSC function could be impaired in a pre-clinical setting that mirrored real world clinical scenarios aimed at achieving TFR – in other words, exposure to a TKI first, followed by addition of an MDM2i, then treatment discontinuation. Using our PDX paradigm, mice were first exposed to NIL for 7–14 days prior to the addition of IDASA for 5 days followed by a 28 day treatment-free period (Fig. 5a). Aligning with our previous results, the LSC (CD34⁺CD38⁻) showed marked decline only in the mice treated with both NIL and IDASA following the 28 day treatment-free period compared to mice treated with NIL only (Supplementary Fig. 6a, b). Strikingly, mice treated with NIL only, showed a significant increase in levels of the most primitive LSC (CD45⁺CD34⁺CD38⁻CD90⁺) at the end of the treatment-free period, when compared to the end of treatment, whilst levels of these primitive cells in mice exposed to NIL and IDASA in combination, declined (Fig. 5b). This was suggestive of the emergence of disease recurrence in the NIL only treated mice, following treatment discontinuation in this arm, but not in mice treated with both NIL and IDASA.

To understand the molecular events that accompanied these observations, we performed scRNA-seq of human CD34⁺ cells engrafted in the BM of treated and untreated mice. Using our knowledge of how the expression of the ESC-REG is altered between cycling and quiescent LSC (Fig. 2c), we identified quiescent LSC as those CD34⁺ cells having low mean expression of the ESC-REG and low expression of *MKI67*, *CD38* and *TP53* (Fig. 5c). The majority of drug-naïve (VEH) quiescent LSC also displayed low levels of mitochondrial (mt) gene expression (Fig. 5d). In contrast, a large proportion of quiescent LSC isolated from mice exposed to NIL plus IDASA displayed high levels of mt gene expression (Fig. 5d; Supplementary Fig. 6c). We attributed this effect to stress and not reduced viability[34], as nuclear gene content was no different between cells with high mt gene expression compared to those with low mt gene expression (Supplementary Fig. 6d). In line with this view, differentially expressed gene (DEG) analysis between the three experimental arms at the end of treatment revealed that quiescent LSC exposed to NIL plus IDASA had significant deregulation of genes modulated during oxidative stress. This included down-regulation of *FOS*, *DUSP1*, and *ZFP36*, the expression of which are critical for LSC survival in response to TKI[35] (Fig. 5e; Supplementary Data 6). In addition, high levels of reactive oxygen species (ROS) in primitive human cells treated with NIL plus IDASA, detected across

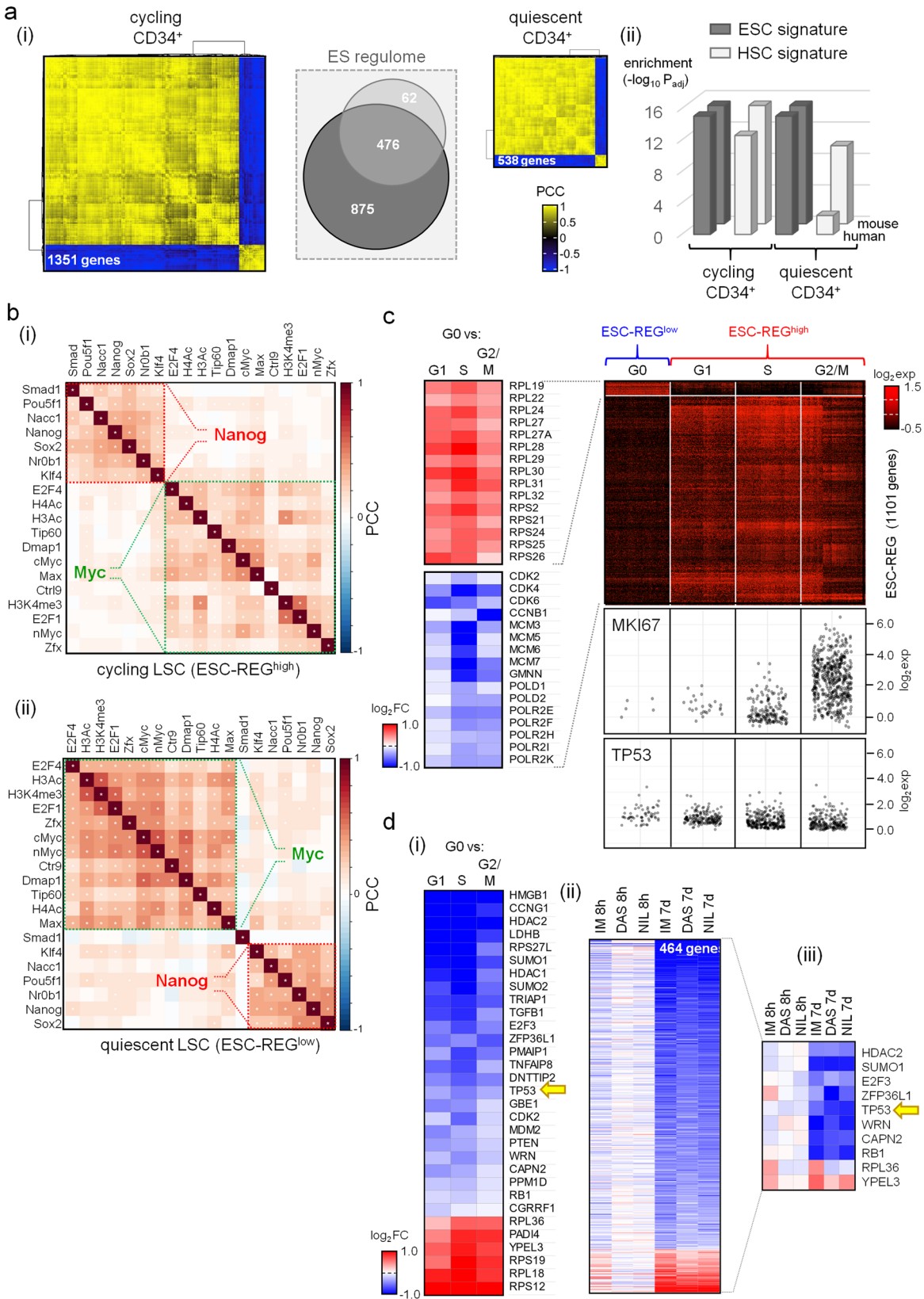

multiple PDX experiments by flow cytometry, reflected the molecular oxidative stress signature (Fig. 5f; Supplementary Fig. 6e). At the end of the 28-day treatment-free period, however, the oxidative stress signature was replaced with a molecular signature of erythroid-like cells (increased haemoglobin gene expression) (Fig. 5g; Supplementary Data 6), suggesting that the oxidative stress response was transient.

## Activation of p53 is accompanied by loss of LSC multi-lineage potential

Given that high ROS levels can lead to cellular commitment and differentiation, we next asked whether p53 activation resulted in changes in lineage potentialities of LSC in our PDX model. We used scRNA-seq to analyse the molecular profiles of human CD45+ cells isolated from

**Fig. 2 | Genes within the ESC-REG, including TP53, are transcriptionally modulated in cycling and quiescent LSC and following TKI treatment. a** (i) Heatmaps depicting Pearson correlation coefficients (PCC) resulting from gene-pair analysis of candidate ESC-REG genes in cycling and quiescent primary CML CD34$^+$ cells. Venn diagram shows which of these genes are found in the ESC-REG of both cycling and quiescent CD34$^+$ cells. (ii) Enrichment values (-log$_{10}$ P$_{adj}$) for human and murine ES cell and adult HSC signatures (StemChecker) for highly correlated genes in cycling and quiescent primary CML CD34$^+$ cells from (i). **b** Heatmaps show PCC analysis for predicted TF and epigenetic co-occupancy of ESC-REG genes, highly correlated in cycling LSC (ESC-REG$^{high}$; 1133 genes) or quiescent LSC (ESC-REG$^{low}$; 207 genes). **c** ScRNA-seq analysis of differentially expressed ESC-REG genes in CML CD34$^+$CD38$^-$ LSC. **Righ**t: Heatmap showing the

absolute expression of 1101 ESC-REG genes in each phase of the cell cycle (500 exemplar LSC per phase). Exemplar LSC in each cell cycle phase having detectable levels of expression for *MKI67* and *TP53* are shown below. **Left:** Heatmap showing DEG analysis of ribosomal protein genes (top) and cell cycle genes (bottom) in quiescent (G0) ESC-REG$^{low}$ LSC versus ESC-REG$^{high}$ LSC in G1, S and G2M. **d** (i) Heatmap showing expression of *TP53* (yellow arrow) and p53 target genes found in the ESC-REG in quiescent ESC-REG$^{low}$ LSC versus ESC-REG$^{high}$ LSC in G1, S and G2M. (ii) 464 genes from the ESC-REG that show differential expression when LSC are treated with a TKI for 8 hrs or 7 days (IM imatinib, DAS dasatinib, NIL nilotinib) (E-MTAB-2594)[59]. (iii) Heatmap showing expression of *TP53* (yellow arrow) and p53 target genes found within the 464 gene set shown in (ii). Source data are provided as a Source Data file.

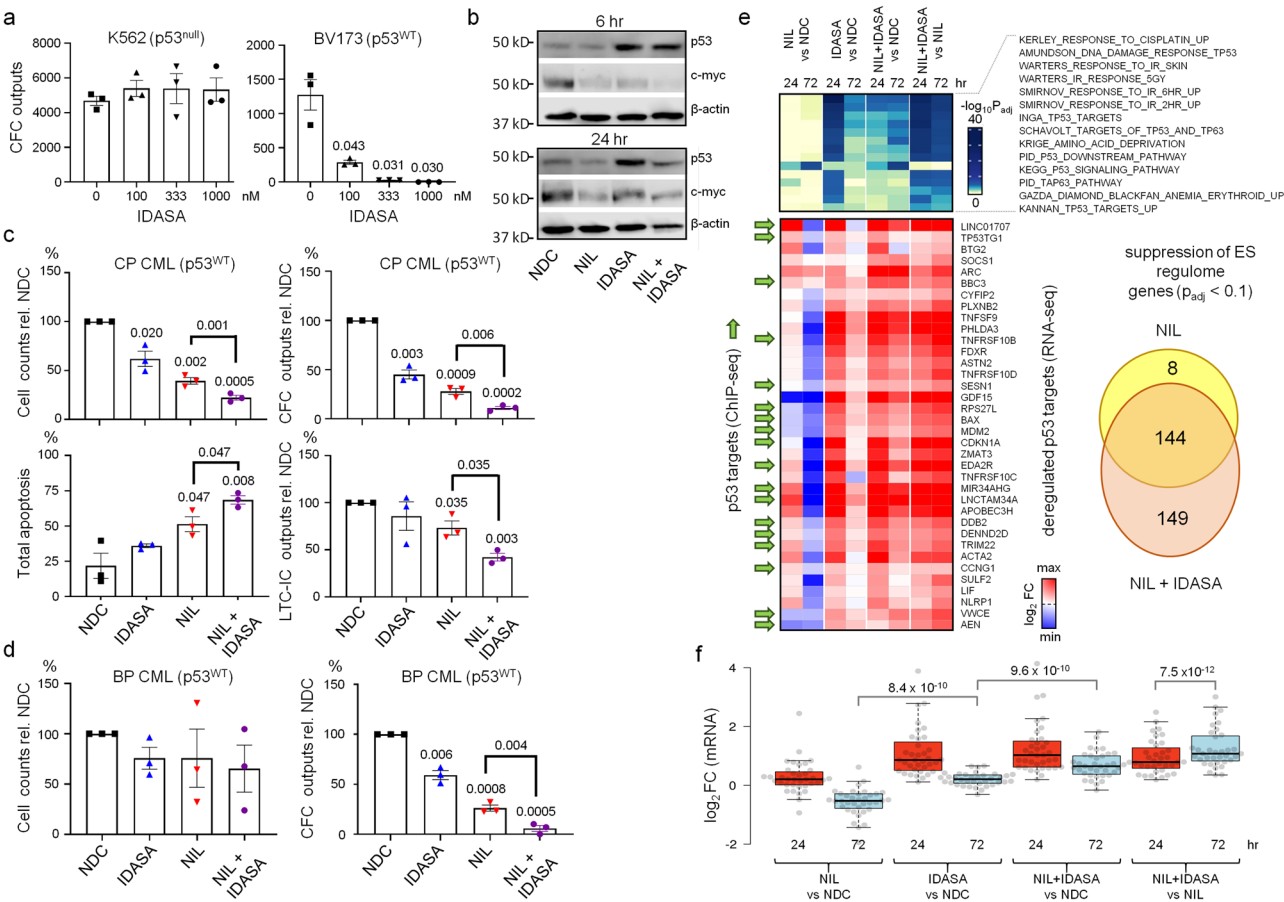

**Fig. 3 | An MDM2 inhibitor, idasanutlin, in combination with nilotinib targets CML stem and progenitor cells in vitro in a p53-dependent manner. a Left:** CFC outputs in K562 cells (p53$^{null}$) after treatment with varying concentrations of ida-sanutlin (IDASA). **Right:** CFC outputs in BV173 cells after treatment with varying concentrations of IDASA (*n* = 3 passages for both). **b** Western blots showing levels of p53 and c-MYC (β-actin, loading control, run on separate blot) when BV173 cells were treated with IDASA (166 nM) +/- nilotinib (NIL;5 nM) for 6 or 24 hrs (replicated 3 times). **c** Effect of treatment with IDASA (100 nM) alone or in combination with NIL (3 μM) in vitro for 72 h on primary CML CD34$^+$ cells (*n* = 3 independent samples) with respect to cell counts, CFC outputs, total apoptosis, and LTC-IC outputs. **d** Effect of treatment with IDASA (300 nM) alone or in combination with NIL (5 μM) in vitro for 72 h on blast crisis primary CML CD34$^+$ cells (*n* = 3 independent samples) with respect to cell counts and CFC outputs. **e** RNA-seq analysis of primary CML CD34$^+$ cells (*n* = 3 independent samples) treated with IDASA (100 nM) alone or in

combination with NIL (3 μM) in vitro for 24 hrs and 72 hrs. Heatmap (top) shows enrichment values (-log$_{10}$ P$_{adj}$) for top-scoring MSigDB signatures between conditions. Heatmap (bottom) showing differential expression (log$_2$ mRNA fold change) of 36 deregulated p53 targets (p53 targets from ChIP-seq = green arrows). Venn diagram shows a total number of ESC-REG genes significantly down-regulated by NIL or NIL + IDASA at 24 hr. **f** Boxplots summarising the changes in mRNA expression of 36 p53 targets (from RNA-seq; *n* = 3 independent samples). *P* values were determined using: two-sided paired Student's t-test (panels **a**, **f**), one-sided paired Student's t-test (panels **c**, **d**), two-sided unpaired Student's t-test (panel **c**, total apoptosis only). Error bars in panels **a**, **c**, **d** are SEM. In panel **f**, center lines show the medians; box limits indicate the 25th and 75th percentiles; whiskers extend 1.5 times the interquartile range from the 25th and 75th percentile; outliers (dots) define minima and maxima. Source data are provided as a Source Data file.

the BM of PDX mice at the end of treatment with VEH or NIL in combination with IDASA. We identified a spectrum of mature human myeloid cell types[36–38] along with stem and progenitor cells displaying ESC-REG circuitry in untreated mice (VEH) (Fig. 6a; Supplementary Fig. 7a, b). However, in mice from the NIL plus IDASA arm, primitive

human ESC-REG cells and those from the neutrophil/monocyte/macrophage (NMM) and megakaryocyte/erythrocyte (ME) lineages were depleted, whilst the eosinophil/basophil/mast cell (EBM) lineage was expanded (Fig. 6a, b; Supplementary Fig. 7a–c). One cell cluster from the NIL plus IDASA arm also showed increased expression of *GATA1*

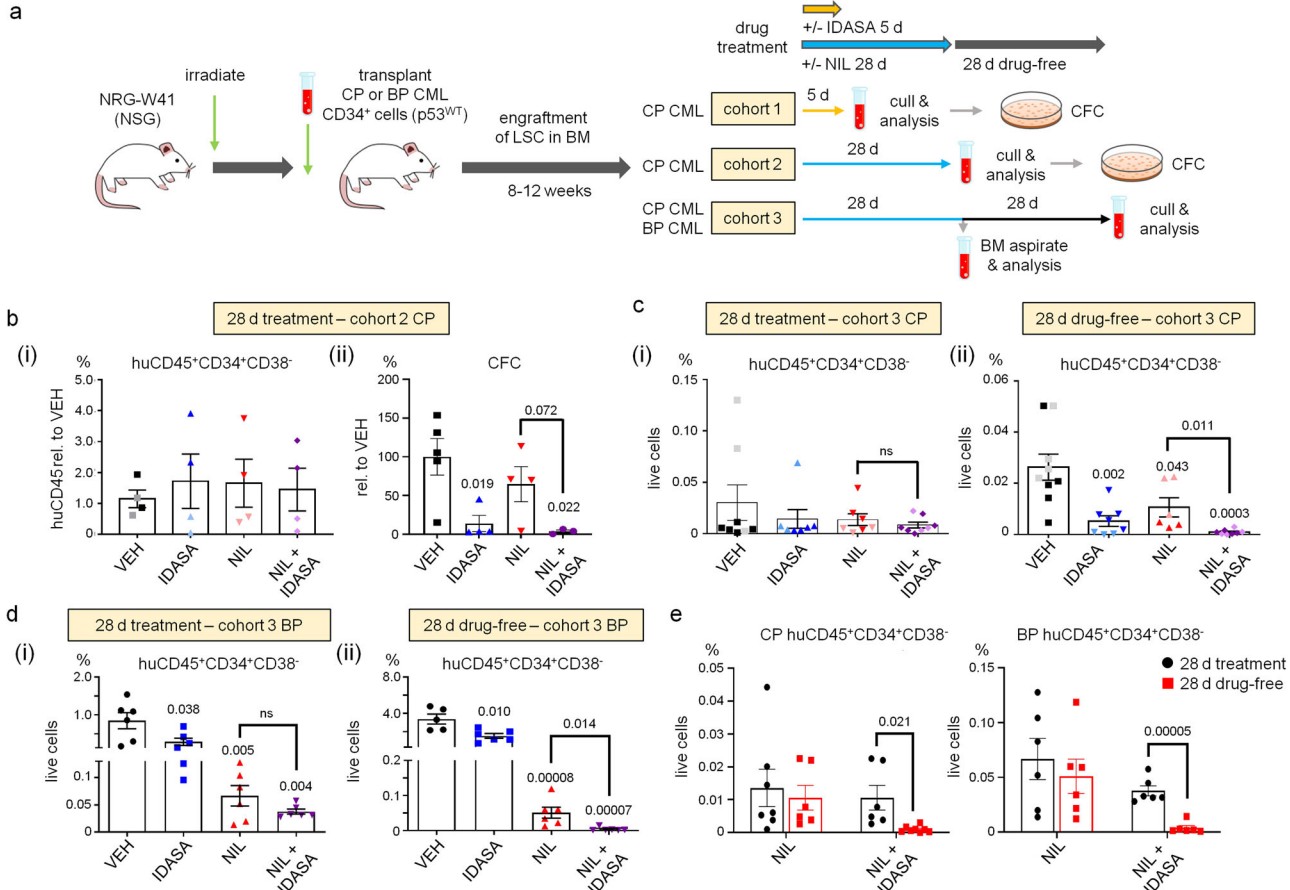

**Fig. 4 | Idasanutlin in combination with nilotinib reduces colony forming capacity at end of treatment, and CD34⁺CD38⁻ CML cells upon drug withdrawal in patient-derived xenografts. a** Experimental design. See also Methods for further details. Mice were treated with nilotinib (NIL) (50 mg/kg QD) for 28 days -/+ idasanutlin (IDASA) (150 mg/kg BD) for the first 5 days of treatment. At the end of 5 days (cohort 1) and 28 days (cohort 2) treatment, mice were analysed for human cell engraftment and clonogenic potential (CFC outputs). A third group of mice had BM aspirates taken for analysis at day 28 and were then left to recover for a further 28 days (treatment-free period) prior to final analysis (cohort 3). **b** Day 28 (cohort 2; chronic phase/CP). (i) Histogram shows levels of huCD45⁺34⁺38⁻ LSC in murine BM expressed as a percentage of huCD45⁺ levels in vehicle-only (VEH) control mice (n = 4/condition); (ii) Histogram shows clonogenic potential expressed as a percentage of the CFC output of VEH per 10,000 huCD45⁺ isolated by flow cytometry

(n = 5 VEH; n = 4 IDASA,NIL; n = 3 NIL + IDASA). **c** Day 28 (cohort 3; CP). Histogram shows the percentage of huCD45⁺34⁺38⁻ cells (i) at end of treatment (n = 8 VEH; n = 7 IDASA, NIL; n = 8 NIL + IDASA) and (ii) at the end of a 28 day treatment-free period (n = 9 VEH; n = 8 IDASA; n = 6 NIL + IDASA) as a percentage of live BM cells. **d** Day 28 (cohort 3; blast phase/BP). Histogram shows the percentage of huCD45⁺34⁺38⁻ cells (i) at end of treatment (n = 6/condition) and (ii) at the end of a 28-day treatment-free period (n = 6/condition) as a percentage of live BM cells. **e** Comparison of the levels of huCD45⁺34⁺38⁻ cells (CP left, BP right) at the end of treatment versus the end of 28 day treatment free period (samples sizes for each condition are as per panels **c** and **d**). In all cases, P values determined using two-sided unpaired Student's t-test; error bars are SEM. Source data are provided as a Source Data file.

(GT1) by DEG analysis, indicative of cells undergoing lineage differentiation[37].

Following on from these observations, we asked whether the molecular signature that we identified in ESC-REG^low LSC at the end of treatment with NIL plus IDASA (Fig. 5e) reflected any molecular rewiring compatible with differentiation. We performed unsupervised clustering of this signature with those obtained by DEG analysis of the untreated and NIL plus IDASA treated human CD45⁺ myeloid cell populations. Strikingly, the end-of-treatment signature of LSC exposed to NIL plus IDASA showed a remarkable similarity with that of the EBM lineage suggesting that these LSC had been re-wired towards an EBM fate (Fig. 6c), and in agreement with the EBM-skewed human CD45⁺ molecular profiles observed in the NIL plus IDASA treated mice (Fig. 6a, b). We confirmed lineage skewing in the human CD45⁺ cells exposed to NIL plus IDASA immunophenotypically by flow cytometry: CD45⁺HLA-DR⁺ cells enriched for monocytes and macrophages were significantly depleted in mice exposed to NIL plus IDASA, whereas levels of CD45^lowSSC^lowCD203c⁺ cells characteristic of basophils were elevated across multiple experiments (Fig. 6d, e; Supplementary

Fig. 7d). We did not observe evidence of such pronounced lineage skewing in human CD45⁺ cells exposed to NIL in any of our PDX experiments, providing support that p53 activation impairs quiescent LSC multi-lineage potential.

## Discussion

We describe here how similarities between the molecular programmes of LSC in CP CML and ES cells can be leveraged to develop a therapeutic strategy to impair LSC function. Such similarities extend beyond mere molecular signatures. Firstly, both ES cells and LSC show increases in the proportion of cells in S and G2/M and a decrease in G0/G1 compared to adult stem cells (Supplementary Fig. 2b). Such reductions in G0/G1 in ES cells are believed to promote pluripotency, discourage lineage commitment and differentiation[29] and be p53-dependent[39]. Secondly, we predict that quiescent LSC share molecular signatures and regulatory programmes with ES cells, based on co-ordinated c-MYC and Nanog circuitry; a regulatory progamme which we demonstrate is BCR::ABL1 dependent (Fig. 1c–f). This circuitry is also reflected in the elevated ribosomal protein gene expression in

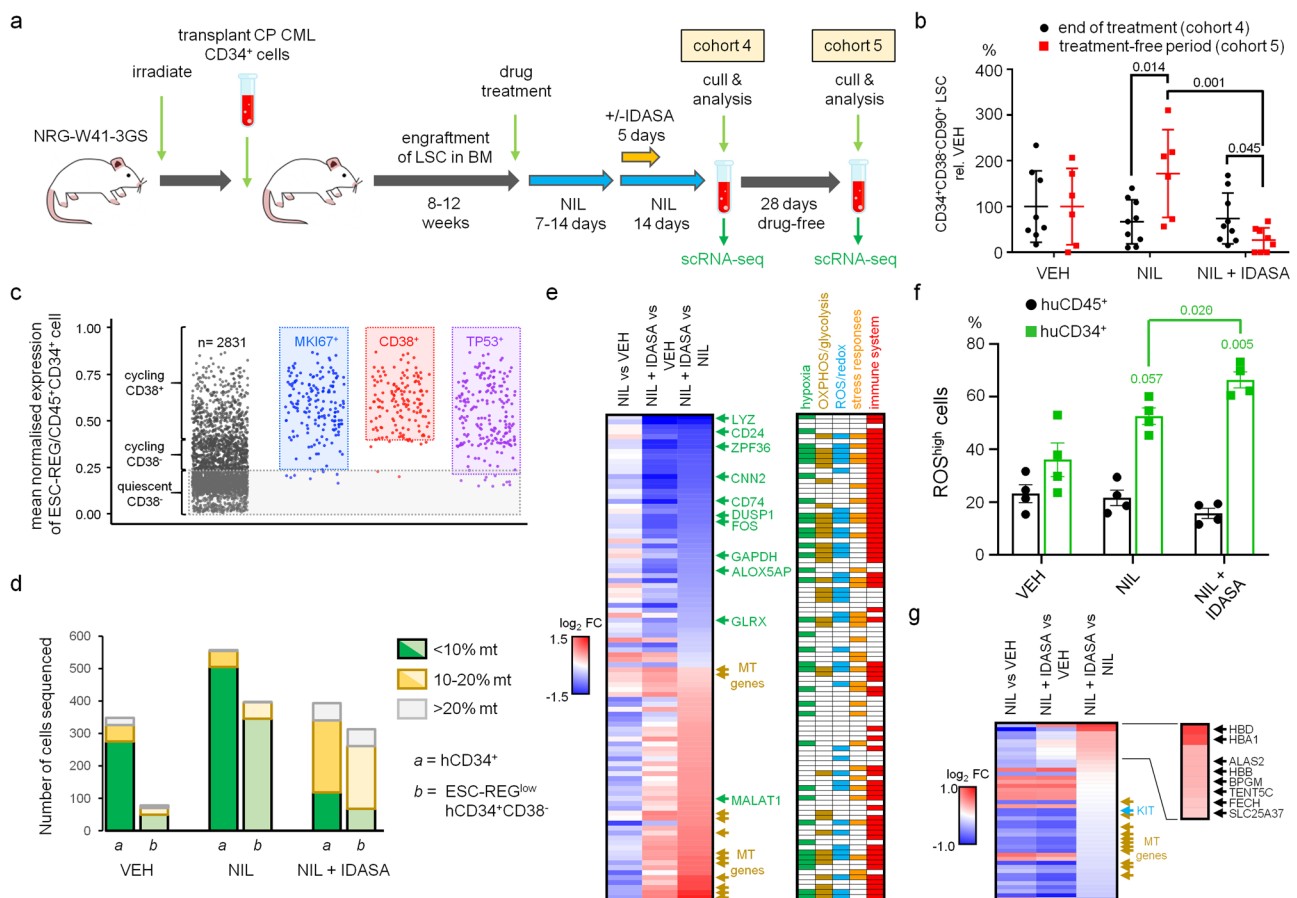

**Fig. 5 | Molecular signatures of quiescent LSC exposed to idasanutlin in combination with nilotinib in vivo. a** Experimental design. See main text and Methods for further details. Mice were treated with nilotinib (NIL) (50 mg/kg QD) for 7–14 days after which the cohorts received NIL for a further 14 days +/- idasanutlin (IDASA) (150 mg/kg BD) for the first 5 days. At the end of treatment (cohort 4) and after a further 28 days treatment-free period (cohort 5) mice were analysed for human cell engraftment and scRNA-seq was performed. **b** Levels of CD34⁺CD38⁻CD90⁺ cells detected at the end of treatment (cohort 4: $n = 8$ VEH, $n = 9$ NIL, $n = 9$ NIL + IDASA) and after a 28 day treatment-free period (cohort 5: $n = 6$ VEH, $n = 6$ NIL, $n = 8$ NIL + IDASA). **c** Dot plots depicting the mean expression of the ESC-REG in huCD45⁺CD34⁺ cells (VEH only from cohort 4; $n = 6$) analysed by scRNA-seq at the end of treatment. Sub-sets of these cells expressing *MIK67*, *CD38*, and *TP53* are shown to the right. Quiescent CD45⁺CD34⁺CD38⁻ LSC (grey panel) lie below the 95th quantile of cells expressing *MKI167*. **d** Histogram showing numbers of

CD45⁺CD34⁺ cells across the three experimental conditions (VEH, NIL, NIL + IDASA) with respect to their percentage of mitochondrial gene content in scRNA-seq (cohort 4; $n = 3$ mice/condition). **e** Heatmap depicting DEG analysis for ESC-REGᶫᵒʷ LSC treated with NIL or NIL + IDASA (cohort 4; $n = 3$ mice/condition). Genes significantly de-regulated by at least +/−0.5 log₂ fold change in one of three comparisons are shown, together with functional annotation (MSigDB; right panel). **f** Histogram showing the percentage of ROSᴴⁱᵍʰhuCD45⁺ or ROSᴴⁱᵍʰhuCD45⁺CD34⁺ cells detected in each experimental condition (end of treatment; $n = 3$ mice/condition; independent experiment from those above). **g** Heatmap depicting DEG analysis for ESC-REGᶫᵒʷ LSC treated with NIL or NIL + IDASA (cohort 5; $n = 3$ mice/condition). Genes significantly de-regulated by at least +/− 0.4 log₂ fold change in one of the three comparisons are shown. *P* values in panels **b** and **f** were determined using the two-sided unpaired Student's t-test; error bars are SEM. Source data are provided as a Source Data file.

quiescent LSC (Fig. 2c), a feature linked to ES cell pluripotency and self-renewal[30]. How LSC mirror ES cells in these ways helps reconcile the phenotypic and molecular similarities previously observed between quiescent and cycling CML LSC[22–24]. In other words, quiescent LSC appear to utilise circuitry normally associated with proliferating ES cells.

Importantly, our data which includes loss of CFC, engraftment and multilineage potentials and reductions in the LSC pool through time as a result of MDM2 inhibition, supports a role for WT p53 in LSC self-renewal - a role we predicted based on the similarities between the molecular circuitry found in ES cells and LSC. This role for p53, to date, is largely unexplored in cancer stem cells where only mutant p53 has been shown to promote self-renewal[40–42]. We demonstrate that increasing p53 levels through MDM2 inhibition, particularly on a background of TKI, has a significant effect on the engraftment potential of LSC and results in loss of multi-lineage potential and LSC exhaustion over time. WT p53 has a well-established role as a negative regulator of self-renewal in adult stem cells, including HSC[43], and in ES

cells and induced pluripotent stem cells (iPSC)[20,21]. In these cells, lowering WT p53 activity facilitates their self-renew. p53 signalling in TKI-naïve CML CD34⁺ cells is indeed lower than that found in normal CD34⁺ cells[18], suggesting LSC self-renewal may be more sensitive to levels of p53 activity than that of HSC, and lowering p53 levels in LSC may be required to balance out the proliferative effects of BCR::ABL1. Low p53 levels in LSC could also explain the similarities between the cell cycle profiles that we observed between ES cells and LSC, given the critical role that p53 has in controlling the ES cell cycle[39].

Defining the ESC-REGᶫᵒʷ and ESC-REGᴴⁱᵍʰ LSC states provides the means to robustly discriminate quiescent from cycling LSC at the molecular level in the scRNA-seq era. Furthermore, the ESC-REG circuitry we describe provides a mechanistic framework through which quiescent LSC, including those exposed to TKI therapy, adopt an ESC-REGᶫᵒʷ state and reduce *TP53* expression, thereby helping to mitigate the negative effects of p53 activity and preserve their ability to self-renew. Given that c-MYC is predicted to be a key modulator of the ESC-REG in CML LSC, and that activation of p53 is associated with down-

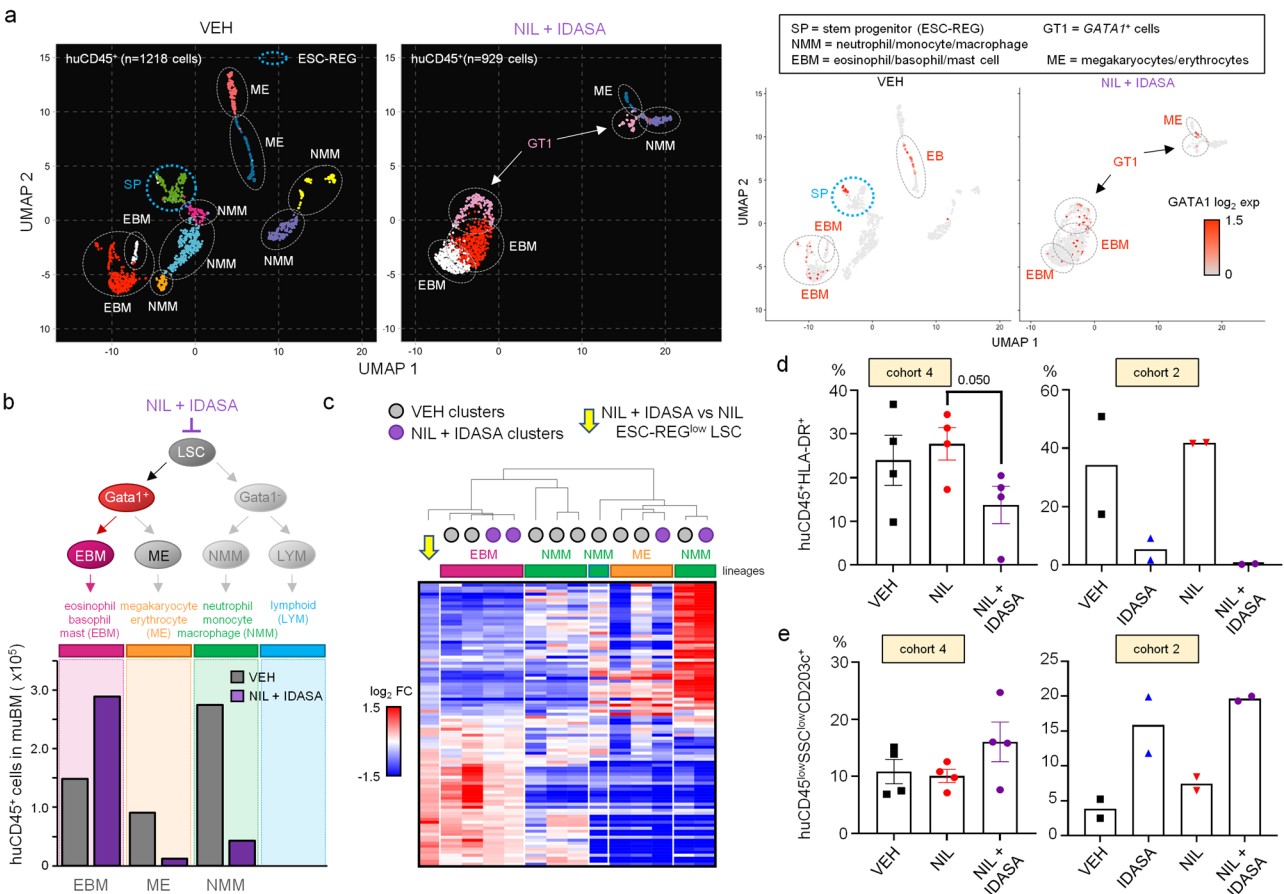

**Fig. 6 | The loss of quiescent ESC-REG^low LSC in vivo results in skewed myeloid outputs. a** scRNA-seq UMAP visualisations of huCD45+ cells isolated from untreated (VEH) mice (n = 1218 cells) and those treated with nilotinib (NIL) + idasanutlin (IDASA) (n = 929 cells) at the end of treatment (PDX; 3 mice/condition; independent experiment to those above) as per experimental design shown in Fig. 5a. Clusters (left panels) are classified by cell type enrichment scores (MSigDB) for up-regulated marker genes, and lineages as defined elsewhere[36,37]. Cluster of stem/progenitor (SP) cells enriched with marker genes found in the ESC-REG is highlighted (blue). *GATA1* expressing cells within these clusters are shown (right panels). **b** Schematic diagram of haemopoietic lineages adapted from[37,38] (top panel). Histogram showing predicted absolute number of human CD45+ cells residing in the BM of mice (PDX) in VEH and NIL + IDASA arms based on scRNA-seq and lineage classifications determined in (**a**) (bottom panel). **c** Heatmap showing unsupervised clustering (Spearman's rank correlation) of the ESC-REG^low LSC oxidative stress gene signature identified at the end of the 28 day treatment (NIL + IDASA vs NIL; yellow arrowhead; see also Fig. 5d) with lineage-defined huCD45+ cell types from UMAP visualisation shown in (**a**). **d** Histograms showing the levels of huCD45+HLA-DR+ cells from PDX where mice have been untreated (VEH) or treated with IDASA, NIL or NIL + IDASA. **e** Histograms showing the levels of huCD45^lowSSC^lowCD203c+ cells from PDX where mice have been untreated (VEH) or treated with IDASA, NIL or NIL + IDASA. Cohorts in (**d**) and (**e**) refer to experiments described in Figs. 4 and 5 (cohort 4, n = 4 mice/condition; cohort 2, n = 2 mice/condition). *P* values shown in (**d**) and (**e**) were determined using the two-sided unpaired Student's t-test. Error bars are SEM. Source data are provided as a Source Data file.

regulation of c-MYC and the ESC-REG (Fig. 3b, e), we cannot overlook the possibility that c-MYC and the ESC-REG also have roles in LSC self-renewal. This is consistent with our previous study which demonstrated that c-MYC has a critical role in LSC survival[18]. Furthermore, we cannot rule out the possibility that induction of apoptosis[33] as well as loss of LSC self-renewal are both consequences of p53 activation in quiescent LSC. We did, indeed, observe induction of apoptosis within 24 hrs of p53 activation in CML CD34+ cells in vitro (Fig. 3c). However, the molecular changes we observed by profiling residual quiescent LSC by scRNA-seq at much later timepoints in vivo, are more characteristic of defects in LSC self-renewal. The transient oxidative stress and high ROS signatures observed in quiescent LSC at the end of treatment in PDX are consistent with deregulated oxidative stress responses being detrimental to self-renewal in HSC[44] and LSC survival[45]. This signature, as well as the erythroid signature identified at the end of the 28 day treatment free period, also reflect changes in cell fate and lineage potentialities within the LSC compartment as a result of p53 activation. Such changes within the quiescent LSC compartment are consistent with findings from other studies which showed that primitive

haemopoietic cells commit to cell fates much earlier than originally believed[37,38,46]. Additionally, inhibition of *ZPF36*, *DUSP1* and *FOS*, three known LSC survival targets[35] may render quiescent LSC sensitive to TKI and contribute to their decline. It also supports a role for these three factors, and others which showed altered expression in quiescent LSC (Fig. 5e) upon combined MDM2 and BCR::ABL1 inhibition, in LSC self-renewal.

Clinically, residual quiescent LSC that persist in virtually all chronic phase CML patients on TKI therapy remains a significant bottleneck to treatment-free remission (TFR) and cure for the overwhelming majority of patients[12]. Achieving very deep responses to TKI therapy have allowed a proportion of chronic phase patients to discontinue TKI and achieve TFR, although mechanisms of how this is achieved are not fully understood[47]. Here we show that activation of p53, in both chronic and blast phases of CML, disables quiescent LSC self-renewal on a TKI background and reduces the level of residual LSC in PDX models - even when TKI treatment is initiated prior to p53 activation. Importantly, the consequence of loss of self-renewal is gradual exhaustion of the residual LSC pool once combined MDM2i

plus nilotinib treatment is discontinued (Fig. 4e). Therefore, based on our data, the introduction of MDM2i into CML clinical trials may represent a highly effective approach to functionally disable LSC, thus preventing them from driving disease recurrence once TKI therapy is discontinued. Such a strategy is compatible with improving rates of TFR for CP CML patients. Given that the overwhelming majority of chronic phase CML patients do not present with, or acquire *TP53* mutations and have WT p53[48,49], p53 activation represents a clinically tractable therapeutic strategy to achieve TFR, with a phase I/2 trial evaluating this strategy currently recruiting (NCT04835584). Some blast phase CML patients however are known to acquire *TP53* mutations, therefore genetic screening of patients would be required before this could be considered a viable treatment for this patient cohort. Given that ES cell circuitry is also found in other cancer stem cells and is likely to regulate multiple pathways and/or cellular functions, further exploration of these functions and developing other therapeutic strategies aimed at deregulating the ESC-REG and self-renewal across cancer types are warranted.

## Methods

### Study approvals
Our study complies with all ethical regulations as follows. Permission to use primary samples from CML patients and normal controls for research purposes was with written informed consent in all cases (West of Scotland Research Ethics Committee 4; REC Ref 10/S0704/2 15/WS/0077, 20/WS/0066). For mouse experiments, all experimental protocols were approved by the local AWERB committee and national Home Office (PD6C67A47), and all methods were carried out in accordance with standard animal housing conditions under local and UK Home Office regulations.

### Primary samples
Primary CML samples (Ph+ in > 90% of CD34+ cells) were obtained from peripheral blood of chronic phase (CP) or blast phase (BP) patients at point of diagnosis (3 bioreplicates from 3 different individuals for all experiments in this study). Normal control primary samples were obtained by mobilisation with granulocyte-colony stimulating factor (G-CSF) during lymphoma staging from individuals with no BM involvement. CML samples were chosen randomly for the experiments described without prior knowledge of age, sex, clinical outcome or response to TKI. The age and sex of individuals at the time of sample collection is provided in Supplementary Table 1. Individuals received no compensation for providing samples.

CD34+ cells were isolated from these samples using CliniMACS (Miltenyi Biotech) and stored frozen in liquid $N_2$. Once thawed, CML CD34+ cells were cultured in serum-free media (20% BIT, 100 μM β-mercaptoethanol, 0.04 mg/mL low-density lipoprotein, 2 mM L-glutamine and 1% penicillin-streptomycin in Iscove's Modified Dulbecco's Medium/IMDM; see Supplementary Data 7 for additional reagent details) containing physiological levels of growth factors (SCF 0.2 ng/mL, G-CSF 1 ng/mL, GM-CSF 0.2 ng/mL, IL-6 1 ng/mL, LIF 0.05 ng/mL, MIP-α 0.2 ng/mL). Control, normal CD34+ cells were cultured in serum-free medium in high growth factors (SCF 100 ng/mL, G-CSF 20 ng/mL, IL-3 20 ng/mL, IL-6 20 ng/mL, Flt3 100 ng/mL). For isolation of stem cells (CD34+CD38−), CD34+ cell samples were stained with anti-human (hu) CD34-APC, CD38-V450 or CD38-Pe-Cy7 antibodies (see Supplementary Data 8 for antibody details), before being subjected to fluorescence-activated cell sorting (FACS) to obtain the stem cell populations using a FACSAria™ Fusion cell sorter (BD Biosciences).

### Cell lines
BV173[50], K562[51], and KCL-22[52] cell lines (mycoplasma-negative; tested in-house on a regular basis) were obtained from DSMZ (cat. no. ACC 20, ACC 10, ACC 519 respectively) and authenticated by Northgene Ltd, Newcastle in 2020 using short-tandem repeat (STR) profiling. Cells were grown in Roswell Park Memorial Institute (RPMI) 1640 medium (Thermo Fischer) supplemented with 10–20% foetal bovine serum (FBS) (Thermo Fischer), 2 mM L-glutamine and 1% (v/v) penicillin-streptomycin. These cells were treated with nilotinib and/or idasanutlin (reconstituted in DMSO) as described in the main text and below.

### Cell expansion, apoptosis, and cell cycle analysis
Primary CD34+ cells from CP or BP CML patients and normal controls were cultured as above for up to three days at concentrations of nilotinib (Selleckchem) and/or idasanutlin (Roche, New York) described in figure legends or main text. Cell expansion over time was determined by counting viable cell numbers using trypan blue exclusion, results are shown relative to the untreated control. To determine the extent of cells moving into apoptosis following compound treatments, cells were incubated with Annexin V–APC and 4',6-diamidino-2-phenylindole (DAPI; Abcam) and analysed by flow cytometry. Annexin V and Annexin V-DAPI positive cells were defined as apoptotic. BV173, K562, and KCL-22 cells were treated with varying concentrations of idasanutlin for 72 hrs. BV173 cells were also treated with nilotinib (5 nM) or idasanutlin (166 nM) or a combination of the two for 6 or 24 hrs.

### Colony-forming cell assays and long-term colony-initiating culture (LTC-IC) assays
For colony-forming cell (CFC) assays, cells cultured with or without idasanutlin ± nilotinib for 3 days were seeded at 1000 cells/mL in 3 mL MethoCult (StemCell Technologies), divided between, and spread over, two 35 mm dishes, and the number and morphology of colonies determined after 12 days incubation at 37 °C and 5% $CO_2$. Alternatively, CFC assays for human CML cells engrafted into the BM of PDX mice and treated with compounds (described below) were carried out by plating $2.5 \times 10^4$ to $1.5 \times 10^5$ human CD45+ cells or $2.5 \times 10^3$ human CD34+ cells divided between two dishes of methylcellulose, incubated and scored as above.

To determine the survival of the more primitive haemopoietic and leukaemic stem cells, long term colony-initiating cell (LTC-IC) assays were used. CML and normal CD34+ cells ($n = 3$ bioreplicates for CML; $n = 3$ bioreplicates for normal) were cultured as above -/+ idasanutlin and/or nilotinib for 3 days before being plated on established M2-10B4 and SL/SL modified murine fibroblast feeder layers. The feeder layer had previously been treated with mitomycin C at 50 μg/mL for 30 min at 37 °C and 5% $CO_2$ to arrest cell cycle progression, then washed thrice with PBS and left in 1 mL MyeloCult (StemCell Technologies) for 24 h. Surviving cells from the drug treatments ($5 \times 10^4$) were placed on the feeder layers in 1 mL MyeloCult in duplicate and maintained for 6 weeks, without the drug and with half medium changes weekly, supplemented with hydrocortisone at a final concentration of $1 \times 10^{-6}$M. After 6 weeks, cells were counted by trypan blue dye exclusion, and all viable cells were transferred to CFC assays in duplicate, maintained in culture for 12–14 days, and then colonies were counted as described above.

### Western blotting
Total cellular protein was extracted from BV173 cells by lysis in RIPA buffer (Thermo Fisher) with the addition of phosphatase and protease inhibitors (both Roche). Cell lysates were run on NuPAGE™ 4–12% Bis-Tris polyacrylamide gels (Thermo Fisher), and transferred to nitrocellulose membrane before being probed with the appropriate antibodies. Antibodies used were p53 (Santa Cruz Biotechnology), c-Myc (Abcam), and b-actin (Cell Signalling Technology) (see Supplementary Data 8). Full scan blots can be found in the Source Data file.

### CML double transgenic (DTG) mouse model
Eight to twelve-week-old B6.SJL recipient mice of both sexes (CD45.1) were irradiated with two doses of 4.25 Gy, 3hrs apart and transplanted

with $2 \times 10^6$ SCLtTA/BCR-ABL (DTG) mouse BM cells (CD45.2) via tail vein injection. The DTG donor cells were derived from a pool of BM from at least ten individual DTG mice of both sexes (8–15 weeks) which had been frozen and stored in liquid $N_2$. Donor cells were recovered from frozen overnight in IMDM + 20% FBS before transplant. Tetracycline (TET) was included in the drinking water for two weeks post-transplant to allow time for recovery and engraftment. Two weeks post-irradiation, TET was removed, allowing BCR::ABL1 expression and leukaemia development. Peripheral blood was obtained from mice by tail vein bleed into EDTA tubes over a 2–3 week period. Following ACK buffer lysis, engraftment and myeloid cell levels were monitored in the blood by staining with antibodies for CD45.1, CD45.2, Gr-1 and Mac-1 followed by flow cytometry to confirm leukaemia development. Mice were treated with vehicle(s), nilotinib (LC Laboratories), RG7112 (Roche, New York) alone or in combination with nilotinib for 28 days. Nilotinib was administered once daily by oral gavage at a dose of 50 mg/kg in its vehicle (10% NMP in PEG300). RG7112 was administered once daily by oral gavage at a dose of 100 mg/kg in its vehicle (2% Klucel LF, 0.10% Tween 80, 0.09% methylparaben, 0.01% propylparaben in water pH 6.46). At the end of treatments, all mice were euthanised and their spleens, blood, and BM harvested. Blood cell counts were analysed using a HemaVet 950 (Drew Scientific), including white blood cells (WBC) and neutrophils. Myeloid cell levels were determined by flow cytometry as above. Spleens were weighed and cells were extracted in PBS/2% FBS. BM was recovered from the hind legs (ilia, femurs, and tibia) of each mouse by crushing using a mortar and pestle in PBS/2%FBS. Cells from the spleen and BM were filtered through 70 mm cell strainers for cell counts and flow cytometry. Cell counts were determined using a Casy cell counter (Cambridge Bioscience), following the manufacturer's instructions. Donor cell engraftment and myeloid cell levels were measured in BM by flow cytometry as above. The following donor (CD45.2) cell populations: LSK (Lin⁻Sca1⁺c-Kit⁺) and LT-HSC (Lin⁻Sca1⁺c-Kit⁺CD150⁺CD48were stained and monitored by flow cytometry using the FACSVerse Flow Cytometer (BD Biosciences) (Supplementary Fig. 8). For secondary transplantation experiments, at end of treatment, BM from the nilotinib ($n = 6$) and RG7112+nilotinib ($n = 5$) treated arms were pooled and $3 \times 10^6$ cells/mouse transplanted into 8–12 week old B6.SJL CD45.1 mice irradiated with $2 \times 400$ cGy, 3 h apart. CD45.2+ cell engraftment levels were monitored in the blood every 4 weeks for 16 weeks at which point the mice were euthanised and multi-lineage engraftment determined in the BM as above. Number of mice used in each DTG experiment were as follows: primary transplants: $n = 44$; secondary transplants: $n = 14$. Mice were monitored closely throughout each experiment for signs of ill-health, distress or leukaemia development (e.g. ruffling of fur, reluctance to move, weight loss, palpable tumours). Mice were culled humanely if body weight loss reached 20% of its starting body weight or when any palpable tumour size exceeded 1.3 cm. In no instances were these limits exceeded. All transgenic mice were kept under the following conditions: light/darkness: 12/12 hr, temperature: 22 + \−2; humidity: 45–65%.

## PDX experiments

CML CD34⁺ cells were transplanted ($1 \times 10^6$ cells per mouse) via tail vein injection into female 8–12-week-old sub-lethally irradiated immune-compromised mice (200 cGy). Male mice were not used as they are more likely to fail to engraft human CML cells and would not contribute data to the study. Mouse models used were NSG (NOD.Cg-*Prkdc^scid Il2rg^{tm1Wjl}*/SzJ]), NRG- *W^{41}/W^{41}* (NOD.Cg-*Rag1^{tm1Mom} Kit^{W-41J} Il2rg^{tm1Wjl}*/EavJ) mice[53] and NRG- *W^{41}/W^{41}* -SGM3 (NOD.Cg-*Rag1^{tm1Mom} Kit^{W-41J} Il2rg^{tm1Wjl}* Tg(CMV-IL3, CSF2, KITLG)1Eav/J) (The Jackson Laboratory, the Eaves laboratory, Vancouver or bred in house). Eight to 12 weeks post-transplant, human cell engraftment was determined by measuring human CD45⁺ cells in the blood of transplanted mice by flow cytometry. Those mice showing evidence of huCD45⁺ cell

engraftment were randomized and treated with vehicle(s), nilotinib (LC Laboratories), and/or idasanutlin according to the schedules described in the main text. Nilotinib was administered once daily by oral gavage at a dose of 50 mg/kg in its vehicle (10% NMP in PEG300). Idasanutlin was administered twice daily by oral gavage at a dose of 150 mg/kg in its vehicle (2% Klucel LF, 0.10% Tween 80, 0.09% methylparaben, 0.01% propylparaben in water pH 6.46). At the end of treatment, all mice were euthanised and the BM harvested from the two hind legs (ilia, femurs, and tibias) of each mouse by crushing in a mortar and pestle as described above. BM was filtered and stained with mouse CD45-APC-Cy7, huCD45-PE, huCD34-APC and huCD38-Pe-Cy7 (all BD Biosciences) and the immunophenotypes of engrafted human cells determined by flow cytometry. For isolation of human cells from PDX model (huCD45⁺ and huCD45⁺34⁺) for scRNA-seq, CFC or FISH, mouse BM cells were co-stained with muCD45 APC-Cy7 and huCD45 PE to distinguish mouse and human cells, together with huCD34 APC where appropriate, before being subject to FACS using the FACSAria™ Fusion cell sorter (BD Biosciences) to isolate the specified human populations (Supplementary Fig. 8). Number of mice used in each PDX experiment (see main text for details) were as follows: cohort 1: $n = 19$; cohort 2: $n = 16$; cohort 3 (CP): $n = 31$; cohort 3 (BP): 24; cohort 4: $n = 26$; cohort 5: $n = 20$. Mice were monitored closely throughout each experiment for signs of ill-health, distress or leukaemia development (e.g. ruffling of fur, reluctance to move, weight loss, palpable tumours). Mice were culled humanely if body weight loss reached 20% of its starting body weight or when any palpable tumour size exceeded 1.3 cm. In no instances were these limits exceeded. All immune-compromised mice were kept under the following conditions: light/darkness: 12/12 hr, temperature: 22 + \−2; humidity: 45–65%.

## Fluorescence in situ hybridization (FISH)

Colonies from LTC-IC clonogenic outputs and human CD45⁺ cells from PDX mice were scored for the presence of the Philadelphia chromosome (Ph⁺). FISH was performed with the LS1 BCR-ABL Dual Color, Dual Fusion translocation probe according to the manufacturer's instructions (Vysis). For each LTC-IC, nuclei from at least 10 colonies were scored where possible. One hundred nuclei were scored from BM from each xenograft.

## Bulk-cell RNA-sequencing

Primary CD34⁺ CML cells ($n = 3$) were treated with nilotinib (3 μM), idasanutlin (166 nM), or a combination for 24 hr and 3 days and cells were pelleted and total cellular RNA extracted from a minimum of $1 \times 10^6$ cells in TRIzoL™ Reagent (Thermo Fisher). The concentration of the RNA was measured using the Nanodrop 2000, and the quality and RIN value was determined using the Agilent Bioanalyser RNA Nano kit (Agilent). RNA libraries were then prepared using the TruSeq RNA Library Prep Kit v2 (Illumina) with single indexing following the manufacturers' instructions. For RNA-seq analysis of murine CD45.2Lin⁻Sca⁺Kit⁺ BM cells (see SCLtTA/BCR-ABL/DTG transplants as described above), BCR::ABL1 expression was induced by TET withdrawal for two weeks in one cohort ($n = 6$), while TET was administered in a second cohort for two weeks to suppress BCR::ABL1 expression ($n = 6$; control group). CD45.2Lin⁻Sca⁺Kit⁺ were isolated from the BM of CD45.1 B6.SJL recipient mice by flow sorting, RNA extracted using the RNAeasy Mini Kit Plus (Qiagen) and libraries prepared using the Kapa RNA HyperPrep kit (Roche). Paired-end sequencing was performed by the Glasgow Precision Oncology Laboratory (GPOL; University of Glasgow) on a HiSeq4000 (Illumina) and each library was sequenced to a depth of at least 50 million reads.

## ChiP-sequencing

BV173 cells were cultured for 6 hrs with and without idasanutlin (300 nM). Ten million cells were pelleted and resuspended in FBS-free RPMI media and cross-linked by adding methanol-free

paraformaldehyde (VWR; final conc. 1%) for 15 min with gentle stirring. Cross-linking was quenched by adding glycine [0.125 M]. Cells were then pelleted, washed in PBS and pelleted again. Cells were resuspended in ~1.5 x pellet volume of cell lysis buffer (10 mM Tris-HCl pH 8.0, 10 mM NaCl, 0.2% Igepal CA-630; protease inhibitors) and incubated for 10 min on ice. The cell nuclei were pelleted for 5 min at 4 °C and resuspended in 1.2 mL of nuclear lysis buffer (NLB 50 mM Tris-HCl pH 8.1, 10 mM EDTA, 1% SDS, protease inhibitors) and incubated on ice for 10 min. Volume was increased to 2 mL by adding 0.72 mL of immunoprecipitation (IP) dilution buffer (20 mM Tris-HCl pH 8.0, 150 mM NaCl, 2 mM EDTA 1% Triton X-100, 0.01% SDS, protease inhibitors) and the chromatin was transferred to a 5 mL tube and sheared to a fragment size of 200-500 bp by sonication. The sheared chromatin was diluted with 4.1 mL of IP dilution buffer and pre-cleared by adding 10 μg rabbit IgG (Millipore) for 1 hr followed by incubation with 200 μL of protein G-agarose bead suspension (Roche) for 3 hr. The protein G-agarose beads were pelleted and 1.35 mL of the chromatin was added to 10 μg of p53 antibody (Santa Cruz Biotechnology) for overnight incubation. ChIP was carried out by adding 100 μL protein G-agarose beads suspension for 3 hr. Beads were washed (twice in 20 mM Tris-HCl pH 8.1, 50 mM NaCl, 2 mM EDTA, 1% Triton X-100, 0.01% SDS; once in 10 mM Tris-HCl pH 8.1, 250 mM LiCl, 1 mM EDTA, 1% IGEPAL CA630, 1% deoxycholic acid; twice in 10 mM Tris-HCl 1 mM EDTA pH 8.0). The immune complexes were eluted from the beads (twice in 225 μL of 100 mM NaHCO3, 0.1% SDS). Crosslinks were reversed by adding 0.2 μL of RNase A (10 mg/mL, ICN) and 27 μL of 5 M NaCl to the eluates with incubation at 65 °C for 6 hr and then overnight at 45 °C in the presence of 90 μg proteinase K (Invitrogen). DNA was purified by phenol-chloroform extraction followed by ethanol precipitation and resuspension in 50 μL of water. ChIP-sequencing libraries were generated using the Nextera XT DNA Library Preparation Kit (Illumina) with dual indexing (Nextera XT Index Kit, Illumina) following manufacturers' instructions. Paired-end sequencing was performed by GPOL on a HiSeq4000 and each library was sequenced to a depth of at least 50 million reads.

## scRNA-seq

Single cell capture and cDNA libraries were prepared using the Chromium™ Single Cell 3' and 5' gene expression library and gel bead kits v2 and v1 respectively and the Chromium™ Single Cell A Chip Kit (10X Genomics) according to the manufacturers' instructions. Briefly, human CD45$^+$/CD45$^+$34$^+$ cells from PDX BM or huCD34$^+$CD38$^-$ cells from primary CD34$^+$ samples ($n = 5$ CML; $n = 5$ normal) were isolated by FACS and carefully washed and resuspended to a concentration of 700 cells/μL and loaded onto the Chromium™ Single Cell A Chip (2000–10000 cells) according to manufacturers' instructions. Samples were then processed on a Chromium Controller (10X Genomics) to produce full length, 10X barcoded cDNA. Following cDNA amplification, single indexed gene expression libraries were constructed, again using the manufacturers protocol. Paired-end sequencing was performed by GPOL on a HiSeq4000 and each library was sequenced to a depth of at least 40,000 reads per cell.

## Analysis of scRNA-seq data

ScRNA-seq data were pre-processed using the 10X Genomics Cell Ranger (3.1.0)[54] suite of analysis pipelines. Demultiplexing of raw base call (BCL) files produced by the Illumina Hiseq4000 platform and conversion to FASTQ file formats was performed using the *cellranger mkfastq* pipeline. Alignment of sequencing reads (FASTQ) to the GRCh38 (GENCODE v32/Ensembl 98) human genome assembly was performed using the *cellranger count* pipeline. Seurat (3.2.3)[55] in R (3.6.3) was used to perform QC and all further analysis and exploration of our scRNA-seq datasets. Single cell transcriptomes that had fewer than 500 genes and greater than 10% mitochondrial gene counts in CD34$^+$CD38$^-$ cells isolated from peripheral blood samples were excluded from further analysis. Single human cells isolated from

murine BM (PDX) that had fewer than 200 genes and greater than 20% mitochondrial gene counts were excluded from further analysis (for reasons discussed in the main text).

All scRNA-seq datasets were normalised using "LogNormalize" with scale.factor = 10000 and scaled using "ScaleData". "RunPCA" was used for dimensionality reduction on the scaled scRNA-seq data. We used stem cell signatures[14,25,26] (Supplementary Data 1) as PCA features for dimensionality reduction of scRNA-seq data from CML and normal CD34$^+$CD38$^-$ cells isolated from peripheral blood. Alternatively, we used the top 2000 most highly variable genes as PCA features for dimensionality reduction of CD45$^+$ or CD45$^+$CD34$^+$ human cells isolated from murine BM (PDX). The first 30 principal components from PCA results were selected to construct a K-nearest neighbour (KNN) graph by "FindNeighbors", which were clustered by Louvain algorithm from "FindClusters" thereafter. We used umap from "RunUMAP" to visualise the clustered cells. "FindMarkers" was performed to find the DEG (i.e., marker genes) of a single cluster compared with all other clusters. Cell cycle phase scores were calculated by "CellCycleScoring" based on sets of curated genes known to be expressed at different stages of the cell cycle[56]. This assigned individual cells to different stages of the cell cycle (G0/G1, S or G2/M based on their relative expression levels of curated cell cycle genes.

## ESC-REG construction

We constructed ES cell regulomes (ESC-REG) for murine BM derived BCR::ABL1 $^+$ Lin$^-$Sca$^-$Kit$^-$ cells (GSE242036), human BM derived CD34$^+$CD38$^-$Lin$^-$ BCR::ABL1 $^+$ cells (GSE76312)[28] and quiescent and cycling CML CD34$^+$ BCR::ABL1 $^+$ cells from peripheral blood (E-MTAB-2508)[23] as follows: we identified candidate regulome genes as the intersection of gene sets from relevant MSigDB[57] signatures (shown in Supplementary Fig. 1a, b) with those that were the union of differentially expressed genes of CML versus normal CD34$^+$CD38$^-$ cells (E-MTAB-2581)[19]. Using gene expression values from the relevant datasets we performed gene-pair Pearson correlation analysis. We defined gene membership of the CML ESC-REG as those genes showing a PCC greater than 0.4 across all the gene-pairs tested. For ESC-REG genes found in quiescent and cycling CML CD34$^+$ cells from peripheral blood, we examined enrichment of adult or embryonic stem cell signatures using StemChecker[58] and projected binary (bound/unbound) TF binding profiles from mouse ES cells[17]. For the ESC-REG identified in quiescent and cycling CD34$^+$ cells from peripheral blood, we performed expression analysis of the CML ESC-REG genes within each stage of the cell cycle by identifying 500 exemplar cells for each cell cycle stage (G1, S, G2M, G0) using the approach outlined above[56] and then by computing cell-to-cell Pearson correlations for those cells assigned to each cell cycle stage.

## Bulk-cell RNA-seq and ChIP-seq analysis

Raw sequencing data was pre-processed using the University of Glasgow Galaxy server (https://www.polyomics.gla.ac.uk/galaxy.html) for which quality control was performed by FastQC (0.72) and the reads were trimmed by Trim Galore! (0.4.3.1) using default parameters. For p53 ChIP-sequencing, the trimmed reads were aligned to human reference genome hg38 using Bowtie2 (2.3.4.2). Narrow peaks were called by MACS2 (2.2.7.1) with q value 0.05. Gene annotation of p53 ChIP-seq peaks employed ChIPseeker (1.28.3) (in R 3.6.3) with the annotation database TxDb.Hsapiens.UCSC.hg38.knownGene. For bulk-RNA-sequencing, the trimmed reads were aligned to human reference genome hg38 using HISAT2 (2.1.0) and assigned by the feature counting tool featureCount (1.6.0.3). DEG were obtained by DESeq2 (1.32.0) in R. Enrichment of molecular signatures in DEG identified from bulk- and single-cell RNA-seq and other transcriptomics datasets utilised MSigDB[57] (v7.4.1) and enrichment was calculated using *enricher* from R package clusterProfiler(4.0.5) with minGSSize=1, and maxGSSize=NA which means no gene set size

restriction. Additionally, we used R package EGSEA (1.20.0) for the signature enrichment analysis for bulk RNA-seq data.

## Statistical analysis of in vitro and in vivo phenotypic data

No datasets were excluded from the analysis and investigators were not blinded to the experimental conditions (for all in vitro assays). Statistical analyses of differences in cell expansion, CFC counts and apoptosis between experimental conditions (as described above) were performed using the one-sided or two-sided paired Student's t-test as appropriate. Statistical analysis of differences in human (PDX) or murine (DTG model) cell populations between experimental conditions (as described above) were performed using the two-sided unpaired Student's t-test.

## Reporting summary

Further information on research design is available in the Nature Portfolio Reporting Summary linked to this article.

## Data availability

The expression profiling RNA-seq data generated in this study have been deposited in the Gene Expression Omnibus (GEO) database and is freely available under accession codes GSE218183, GSE218184, GSE218185, and GSE242036. The ChIP-seq dataset generated in this study has been deposited in the GEO database and is freely available under accession code GSE218182. The publicly available datasets used in this study are available in the GEO database under accession code GSE76312[28] and in the EMBL-EBI database under accessions E-MTAB-2508[23], E-MTAB-2581[19] and E-MTAB-2594[59]. Additional information concerning human samples can be obtained from the corresponding author. The remaining data are available within the Article, Supplementary Information or Source Data file. Source data are provided with this paper.

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

## Acknowledgements

The authors thank all those individuals with CML who generously provided samples, and the technical support provided by the sequencing facility at the Glasgow Precision Oncology Laboratory (GPOL; University of Glasgow) and the animal unit at the CRUK Scotland Institute. This study was supported by Blood Cancer UK (Refs. 14033 and 21004; DV, MC), the Stand Up To Cancer campaign for Cancer Research UK (Ref. C55731/A24896; DV, MC), the Howat Foundation (MC), the Glasgow Experimental Cancer Medicine Centre (funded by Cancer Research UK and the Chief Scientist's Office, Scotland; MC), and Najran University (Saudi Arabia; HA).

## Author contributions

Conceptualised, designed and supervised the study: M.T.S., W.L., H.J., M.C., and D.V.; performed in vitro experiments: F.W., R.M., J.P.; performed bulk- and scRNA-sequencing: M.T.S., H.A., C.C.; performed in vivo experiments: M.T.S., R.M., C.C., K.D., M.E.D., J.P., A.M.M.; performed ChIP-sequencing: R.K., D.V.; curated and provided patient samples: H.J., M.C.; analysed experimental data: M.T.S., W.L., R.M., C.C., T.S., F.W., R.K., J.P., D.V.; developed/provided MDM2 inhibitors at Roche: B.H.; wrote the draft manuscript: M.T.S., W.L., D.V.; reviewed & edited the manuscript: M.T.S., W.L., R.M., T.S., H.J., A.M.M., B.H., M.C., D.V.; acquired funding: M.C., D.V. M.T.S. drove analysis performed by W.L. and T.S., authorship was assigned accordingly.

## Competing interests

M.C. has received research funding from Cyclacel and Incyte, honoraria from Astellas, Novartis, Incyte, Pfizer, and Jazz Pharmaceuticals, and is/has been an advisory board member for Novartis, Incyte, Jazz Pharmaceuticals and Pfizer. B.H. is an employee of Genentech Inc (Roche Group). The remaining authors have declared that no conflict of interest exists.
