## [Peer Review File · Nature Communications]

Reviewers' Comments:

Reviewer #1:

Remarks to the Author:

This paper reports the results of extensive omic measurements including multiple single-cell RNA-seq analyses of purified CD34+ phenotypes of BCR-ABL+ chronic phase patients' samples obtained at diagnosis before and after exposure to nilotinib and a known inhibitor of MDM2 (idasanutlin) in vitro and in engrafted immunodeficient mice that together support their hypothesis that the cells responsible for maintaining the chronic phase clone express low levels of P53 and, if this is reversed, these cells become susceptible to being eliminated by nilotinib and hence eliminate the chronic phase population.

Strengths: The experiments appear carefully performed and the results carefully analyzed on highly relevant primary patients' cells in state of the art in vitro and in vivo protocols with added support from experiments with the best mouse model available. The treatment designs appear carefully chosen to serve as a precursor to a potential clinical trial and provide results that would support such a next step.

Weaknesses:

1. The avenue of discovery from the lead comparison to ESC maintenance circuitry is not convincing per se except to lead to an approach to manipulate CML cells that looks promising. Moreover, the subsequent experiments that generate results consistent with the proposed effects do not formally test the inferred mechanisms which in a much larger population with other confounding mutations or given other TKIs remains untested. This is not mandatory but the manuscript would benefit from better acknowledgement of these caveats and perhaps not give such weight to the observations that led to the treatment designs used.
2. In addition there is a confused use of molecular and biological terminology. For example, "LSCs" are defined in the Methods phenotypically and then effects of various treatments on them discussed in terms of effects on CFC maintenance and secondary transplant capabilities that are indicators but not quantitative and hence difficult to extrapolate to anticipated effects on a adult person. This should be addressed by using explicit terms for endpoints measured and their quantitative limitations, and avoiding biological terms for drawing inferences from different measurements of unknown quantitative relationships.
3. The results would also benefit being presented in a more succinct form without such extensive rationalisation and implied import given their limited correlative nature and lack of additional mechanistic experiments. In this regard, the implied mechanistic relevance of induced oxidative stress is another example of an interesting speculative finding without substantive follow-up and might be better included in a more concise earlier description of the inferences of the omic findings and their consistencies with the later experimental outcomes handled in the Discussion.

Minor

1. Page 4, line97 – what is the molecular or biological meaning of "propensity" in this context. Please make more precise.
2. Page 5, line 105 –self-renewal is not a clearly defined term here so claims of "first time" seem out of place.
3. Page 5, line 118 – please add a reference for "DREAM complex targets"
4. Page 6, line 136 – "defined" in this context seems too strong.
5. Page 6, lines 141-142 – This is an overstated extrapolation undermined by the use of undefined terms.
6. Page 8, lines 186-188 and Page 11 lines 252 271 – again examples of lack of precision of exactly what is self-renewal in terms of a quantifiable entity at the single cell level based on a confusion of the results expected or attained and a mechanistic cause of what was actually measured.

Reviewer #2:

Remarks to the Author:

This work has merit because limiting the duration of TKI therapy is an important goal in treating patients with the disease. The authors pose an interesting hypothesis whereby combination of a TKI with an MDM2 inhibitor results in an embryonic stem cell signature that impairs clonogenicity,

thus retarding the fitness of the CML clone.

CD34+ CD38- CML LSC samples were obtained from PB. Could the authors outline whether these have the same gene expression profile as progenitors obtained from the bone marrow?

The control HSC population were derived from G-CSF stimulated cell populations. Are there any confounding effects possible from cytokine stimulation that could have influenced the transcript profiles described?

Please define the term SSC

What do the authors propose is driving the ESC signature? Is this BCR-ABL dependent? If this fusion is transduced into normal progenitors, is the ESC transcription signature reproduced?

Is this the same signature present in leukaemic progenitors from Philadelphia positive ALL cases?

Another paper suggests that CML leukemic stem cells are characterized by high levels of TP53. <https://doi.org/10.3324/haematol.2019.219261>

Please comment on what might explain the discordance with the current findings.

In Fig 3B, the % of CD34+CD38- negative cells are very low making it difficult to observe statistical differences within the various treatment arms.

In Fig 3Bii, is there a significant difference between the two IDASA arms and vehicle? Does this suggest some progenitor toxicity resulting from the MDM2 inhibitor? How then do we separate reduced LSC numbers from a reduction in self renewal properties? Could this just reflect fewer viable cells from combination drug therapy? Relative % may not easily reflect an absolute reduction in the number of cells when the denominator is so large.

Is the TKI induction of ESC-REG genes reproducible in BCR-ABL models with TP53 KO? Does this align with the author's proposal that TKI induces ESC-REG genes despite apparently not inducing TP53 itself?

Are IDASA treated LSCs reduced by the toxic effects of the drug rather than just by a loss of clonogenic function? Is the reduced CFC output also observed in TP53 KO settings?

To control for how important it is for the treatment free period to impair CFC function, are the CFC outputs higher in patients remaining on NIL continuously rather than with the 28-day break?

It is quite hard to interpret what is going on in figure 3. It may be helpful to have a table listing exactly what treatments were provided in each arm.

Although these observations are interesting, is there any evidence of a long-term survival benefit in these models from the combination of IDASA with NIL?

Are the authors suggesting that just one five day treatment will be sufficient to improve survival in CML or would IDASA cycles need to be delivered recurrently?

If recurrently, is there risk from prolonged use of an MDM2 inhibitor that a more resistant p53 mutant clone will be selected?

Reviewer #3:

Remarks to the Author:

This paper is a mix of gene expression analysis, cell biology in vitro and in vivo experiments. For anyone less expert than the authors in the clinical problem and the lab approaches it is overly complex overly detailed and very hard to follow. One has to have faith that the ESC like profiles used to compare HSCs and LSCs are robust. There is no direct comparison with ESC in the paper.

Why at the outset were peripheral blood LSCs used. Are these cells representative of those that drive relapse on discontinuation of TKIs? It is claimed that the key activity of MDM2-inhibition, that underpins the translational argument for patient benefit and increased therapy free remission, is p53 activation. However, there are no assays of p53 activation and no consideration of other off-target effects of IDAS treatment that may be key to the effects of dis-empowering LSCs.

Sept 1, 2023

Ref: NCOMMS-22-43264-T

Rebuttal to reviewers.

Reviewer #1 - CML LSCs, P53 (Remarks to the Author):

This paper reports the results of extensive omic measurements including multiple single-cell RNA-seq analyses of purified CD34+ phenotypes of BCR-ABL+ chronic phase patients' samples obtained at diagnosis before and after exposure to nilotinib and a known inhibitor of MDM2 (idasanutlin) in vitro and in engrafted immunodeficient mice that together support their hypothesis that the cells responsible for maintaining the chronic phase clone express low levels of P53 and, if this is reversed, these cells become susceptible to being eliminated by nilotinib and hence eliminate the chronic phase population.

Strengths: The experiments appear carefully performed and the results carefully analyzed on highly relevant primary patients' cells in state of the art in vitro and in vivo protocols with added support from experiments with the best mouse model available. The treatment designs appear carefully chosen to serve as a precursor to a potential clinical trial and provide results that would support such a next step.

R. We thank the reviewer for highlighting the strengths of our manuscript.

Weaknesses:

1. The avenue of discovery from the lead comparison to ESC maintenance circuitry is not convincing per se except to lead to an approach to manipulate CML cells that looks promising. Moreover, the subsequent experiments that generate results consistent with the proposed effects do not formally test the inferred mechanisms which in a much larger population with other confounding mutations or given other TKIs remains untested. This is not mandatory but the manuscript would benefit from better acknowledgement of these caveats and perhaps not give such weight to the observations that led to the treatment designs used.

R1. We do mention at the end of our discussion, that understanding of the ES cell circuitry in CML warrants further exploration – thus acknowledging the caveats highlighted by the reviewer. Furthermore, we have added additional transcriptomic analysis to demonstrate that the ESC circuitry we initially identified in peripheral blood CD34+CD38- obtained from chronic phase CML patients, is also present in primitive bone marrow cells obtained directly from chronic phase CML patients, and in the LSK cells of our transgenic mouse model of CML-like disease. We also provide evidence that this ES cell circuitry is BCR-ABL1 dependent (Figure 1). These additional analyses strengthen our hypothesis that, as in ES cells, p53 has a role in LSC self renewal, which we subsequently support through the outcomes of our various end point analyses.

The incidence of additional mutations is rare in chronic phase CML. However, these do occur at a higher frequency in blast phase disease. Therefore, to expand the clinical significance of our study, and address the reviewer’s concern, we have now examined the effects of MDM2 inhibition in primary patient samples in blast phase CML (both in vitro and in PDX). We see a similar outcome in these experiments as we show in samples with chronic phase disease (Figure 4). Testing of the validity of our proposed therapeutic strategy would be best done in the context of a trial where patients exposed to a variety of TKI would be recruited. Furthermore, we highlight in the discussion that patients should be pre-screened for p53 mutations to determine their suitability for recruitment to trials where the efficacy of MDM2i are to be evaluated.

2. In addition there is a confused use of molecular and biological terminology. For example, “LSCs” are defined in the Methods phenotypically and then effects of various treatments on them discussed in terms of effects on CFC maintenance and secondary transplant capabilities that are indicators but not quantitative and hence difficult to extrapolate to anticipated effects on a adult person. This should be addressed by using explicit terms for endpoints measured and their quantitative limitations, and avoiding biological terms for drawing inferences from different measurements of unknown quantitative relationships.

R2. We thank the reviewer for raising this point, and we agree that our end point analyses are not direct readouts of self-renewal. However, our end point analyses do measure (quantitatively or qualitatively) biological properties of primitive leukaemic cells having self-renewal capacity (high CFC outputs, engraftment potential, multi-lineage potential, and the ability to maintain the stem cell pool). In our revised manuscript, we have been more careful in our terminology and explicitly state what we are measuring and how it relates to LSC self-renewal at the beginning of the results section entitled “Mdm2 inhibition impairs LSC survival *in vitro* in a p53-dependent manner”.

3. The results would also benefit being presented in a more succinct form without such extensive rationalisation and implied import given their limited correlative nature and lack of additional mechanistic experiments. In this regard, the implied mechanistic relevance of induced oxidative stress is another example of an interesting speculative finding without substantive follow-up and might be better included in a more concise earlier description of the inferences of the omic findings and their consistencies with the later experimental outcomes handled in the Discussion.

R3. We have extensively re-written the first few sections of the results, to provide a more succinct description of the ES cell circuitry (and with additional data included). We have removed references to oxidative stress from the abstract and refer to our molecular data as pointing to “signatures of oxidative stress”, which we discuss in the context of our other data in the discussion section.

Minor

1. Page 4, line 97 – what is the molecular or biological meaning of “propensity” in this context. Please make more precise.

R1. We have removed the term propensity and re-worded this sentence.

2. Page 5, line 105 –self-renewal is not a clearly defined term here so claims of “first time” seem out of place.

R2. We have removed the term “first time” from this sentence.

3. Page 5, line 118 – please add a reference for “DREAM complex targets”

R3. We have included a citation for this term.

4. Page 6, line 136 – “defined” in this context seems too strong.

R4. This phrase is not in our revised manuscript.

5. Page 6, lines 141-142 – This is an overstated extrapolation undermined by the use of undefined terms.

R5. This phrase is not in our revised manuscript.

6. Page 8, lines 186-188 and Page 11 lines 252 271 – again examples of lack of precision of exactly what is self-renewal in terms of a quantifiable entity at the single cell level based on a confusion of the results expected or attained and a mechanistic cause of what was actually measured.

R6. We have removed these imprecise descriptions from our revised manuscript.

Reviewer #2 - Leukaemia, indasanutlin (Remarks to the Author):

This work has merit because limiting the duration of TKI therapy is an important goal in treating patients with the disease. The authors pose an interesting hypothesis whereby combination of a TKI with an MDM2 inhibitor results in an embryonic stem cell signature that impairs clonogenicity, thus retarding the fitness of the CML clone.

R. We thank the reviewer for their comprehensive review. We would, however, like to clarify that we have not stated at any point in our manuscript, that a TKI with an MDM2 inhibitor results in an embryonic stem cell signature. Our molecular data (Figures 3, 5 and 6) points to upregulation of a p53 signature, deregulation of oxidative stress signatures and loss of multilineage potential.

Q1. CD34+ CD38- CML LSC samples were obtained from PB. Could the authors outline whether these have the same gene expression profile as progenitors obtained from the bone marrow?

R1. We and others have used CD34+CD38- CML LSC obtained from peripheral blood to identify molecular features that are drug targetable and that are known drivers of ES cell biology (Myc, EZH2) (see ref 18 and 19). In our revised manuscript we include analysis of primitive bone marrow cells from chronic phase patients and demonstrate that these cells also have a gene expression patterns consistent with an ES cell regulome. We demonstrate the same for murine Lin-Sca+Kit+ cells in which BCR-ABL1 is expressed (both Figure 1). Therefore analysis of both PB and bone marrow cells yield the same outcomes.

Q2. The control HSC population were derived from G-CSF stimulated cell populations. Are there any confounding effects possible from cytokine stimulation that could have influenced the transcript profiles described?

R2. G-CSF mobilise HSC have been used in many published studies to examine the differences between HSC and LSC (including our own). It is known that CML LSC produce autocrine G-CSF which is believed to enable mobilisation of CML LSC to the blood – therefore G-CSF mobilised HSC represent a good control for our analysis. Importantly, we have included new data in Figure 1 which compare primitive human or murine BCR-ABL1 +/- cells from the bone marrow. We hope our responses to both points R1 and R2 reassure the reviewer of the biological relevance of our findings.

Q3. Please define the term SSC

R3. We apologise that this term was inadvertently included throughout the text to refer to adult/somatic stem cells. We no longer use this term (calling them adult stem cells in our revised manuscript). We would also like to note that SSC is also an acronym used in flow cytometry (legend of Supplementary Figure S8).

Q4. What do the authors propose is driving the ESC signature? Is this BCR-ABL dependent? If this fusion is transduced into normal progenitors, is the ESC transcription signature reproduced?

R4. In our revised manuscript, we have included additional data in Figure 1 which demonstrate that the formation of the ESC signature is BCR-ABL1 dependent.

Q5. Is this the same signature present in leukaemic progenitors from Philadelphia positive ALL cases?

R5. Whilst this is an interesting question to raise, the analysis of Ph+ ALL cases is beyond the scope of our study which focusses only on the biology of CML LSC.

Q6. Another paper suggests that CML leukemic stem cells are characterized by high levels of TP53.

<https://doi.org/10.3324/haematol.2019.219261>

Please comment on what might explain the discordance with the current findings.

R6. The published study that the reviewer refers to demonstrated that p53 levels are higher in CML primitive cells compared to normal. We make no such claim in our study. Our analysis focuses on the difference between cycling LSC and quiescent LSC which was not addressed in the previous study.

Q7. In Fig 3B, the % of CD34+CD38- negative cells are very low making it difficult to observe statistical differences within the various treatment arms.

R7. We have included an exemplar FACS gating strategy and real FACs plots from our PDX experiments in Supplementary Figure S8, to reassure the reviewer that we are collecting sufficient events during FACS analysis to accurately quantify even rare primitive CML populations such as CD90+ LSC. Please note that % live cells refers to % of all viable cells in our murine bone marrow – including both mouse and human cells.

Q8. In Fig 3Bii, is there a significant difference between the two IDASA arms and vehicle? Does this suggest some progenitor toxicity resulting from the MDM2 inhibitor? How then do we separate reduced LSC numbers from a reduction in self renewal properties? Could this just reflect fewer viable cells from combination drug therapy? Relative % may not easily reflect an absolute reduction in the number of cells when the denominator is so large.

R8. We agree with the reviewer that because there are reductions in the levels of LSC found in our drug treated arms at day 5, we cannot exclude whether these reductions are the cause of reduced CFC outputs we observe at this time point. We have revised our manuscript and state that only significant reductions in CFC outputs were found at the end of 28 days of treatment – and not after 5 days.

Q9. Is the TKI induction of ESC-REG genes reproducible in BCR-ABL models with TP53 KO?

Does this align with the author's proposal that TKI induces ESC-REG genes despite apparently not inducing TP53 itself?

R9. As stated above (R4) the ESC-REG is BCR-ABL1 dependent. We have not stated anywhere in our manuscript that the ESC-REG is dependent on TP53 expression. Therefore, the experiment suggested by the reviewer, whilst interesting, is beyond the scope of our study. Furthermore, we show data that supports that TKI alone represses the ESC-REG including TP53. TKI does not induce the ESC-REG as suggested by the reviewer.

Q10. Are IDASA treated LSCs reduced by the toxic effects of the drug rather than just by a loss of clonogenic function? Is the reduced CFC output also observed in TP53 KO settings?

R10. In our revised manuscript, we have included data showing that only p53 wild type CML cells show decreased clonogenicity in response to IDASA, while the drug has no effect in CML cells where p53 is mutant/inactive (Figure 3a). This provides strong evidence that the effects we observed with IDASA are not the result of generalised toxicity and are p53 dependent.

Q11. To control for how important it is for the treatment free period to impair CFC function, are the CFC outputs higher in patients remaining on NIL continuously rather than with the 28-day break?

R11. We do not dispute the effectiveness of TKI in controlling CML disease, and acknowledge that maintaining mice on NIL for an additional 28 days would be an

effective treatment. However, as the goal of our study to mimic whether MDM2 inhibitors when added to a TKI could improve the likelihood of TFR, we need to be able to demonstrate what the outcome would be if we discontinued treatment of mice treated with NIL only. We therefore believe that we have used the most suitable controls to conduct these experiments.

Q12. It is quite hard to interpret what is going on in figure 3. It may be helpful to have a table listing exactly what treatments were provided in each arm.

R12. We thank the reviewer for pointing this out. We have revised the schematic diagram to make it easier to understand our experimental paradigms and their endpoints (now shown as Figure 4A in the revised manuscript).

Q13. Although these observations are interesting, is there any evidence of a long-term survival benefit in these models from the combination of IDASA with NIL?

R13. We acknowledge that survival studies are one of the principal endpoints in many translational cancer studies. However, we do not believe survival studies are relevant or feasible for our study for the following reasons:

1. Since the introduction of TKIs, the vast majority of chronic phase CML patients have life spans approaching that of the normal population. Therefore, performing pre-clinical survival studies in chronic phase CML do not mirror real-world clinical scenarios for CML patients, where therapy discontinuation is now the principal goal. Our experimental paradigms, where we withdraw therapy from mice and observe the effects on the LSC pool, are the best approximation to mirror this clinical scenario.
2. Furthermore, it is not feasible to use PDX models of chronic phase CML for survival studies. Chronic phase CML samples engraft at low levels of in murine bone marrow (between 1-10% of total cells). Because of this, these mice do not ever succumb to CML-like disease and would be unsuitable for survival studies.
3. Others (<https://doi.org/10.3324/haematol.2019.219261>) have shown that using an MDM2 inhibitor in combination with a TKI in a transgenic mouse model of CML does indeed provide a long-term survival benefit (and we refer to this study in our manuscript). In fact, this previous study used the same transgenic model we used here in our study. They demonstrated that MDM2i plus a TKI resulted in a survival advantage – but only after these studies were carried out over more than 400 days. When considering the 3R principles of animal welfare, we did not think we should repeat such survival experiments in our study given that strong data for this had already been published.

Q14. Are the authors suggesting that just one five day treatment will be sufficient to improve survival in CML or would IDASA cycles need to be delivered recurrently?

R14. We acknowledge that pre-clinical findings, never truly mirror the real-world (although we have done our best to conduct our experiments to reflect real-world clinical settings). Whether a single cycle of IDASA is sufficient to allow more patients to discontinue therapy, should only be tested in future clinical trials.

Q15. If recurrently, is there risk from prolonged use of an MDM2 inhibitor that a more resistant p53 mutant clone will be selected?

R15. As p53 mutations are rare in chronic phase CML (confirmed with deep sequencing at diagnosis), the likelihood of the emergence of p53 mutant clones in patients is low when exposed to MDM2 inhibitors. However, as we state in the discussion, we would advise that all patients being considered for clinical trials where MDM2 inhibitors are being test, to be pre-screened for p53 mutations, and the outcomes of this be used determine eligibility for trial enrolment.

Reviewer #3 - LSCs, leukaemia therapy (Remarks to the Author):

This paper is is a mix of gene expression analysis, cell biology in vitro and in vivo experiments. For anyone less expert than the authors in the clinical problem and the lab approaches it is overly complex overly detailed and very hard to follow.

R. We have tried to simplify and figures accordingly to make our study easier to follow to the non-expert reader.

Q1. One has to have faith that the ESC like profiles used to compare HSCs and LSCs are robust. There is no direct comparison with ESC in the paper.

R1. The regulatory circuitry of ES cells have been well studied over the last two decades (we cite these seminal studies in our manuscript). Because of this we use the circuitry described in ES cells as a means to identify the Myc and Nanog modules we observed in CML LSC. We therefore have made a direct comparison with ES cells in our study.

Q2. Why at the outset were peripheral blood LSCs used. Are these cells representative of those that drive relapse on discontinuation of TKIs? Non responders....

R2. As described in our response to reviewer 2 (R1), peripheral blood LSC are an appropriate cell source to understand aspects of LSC biology in CML.

Q3. It is claimed that the key activity of MDM2-inhibition, that underpins the translational argument for patient benefit and increased therapy free remission, is p53 activation. However, there are no assays of p53 activation and no consideration of other off-target effects of IDAS treatment that may be key to the effects of dis-empowering LSCs.

R3. We have showed data demonstrating p53 activation (apoptosis assays, western blotting, upregulation of p53 targets in RNA-seq and CHIP-seq data) when IDASA is studied in CML cells where p53 function is wild type, and demonstrated that when p53 is mutant there are no effects on CFC outputs (Figure 3A) demonstrating that off-target effects of IDASA are unlikely to be important in dis-empowering LSC.

Reviewers' Comments:

Reviewer #1:

Remarks to the Author:

I am happy with the revisions made and recommend acceptance.

Reviewer #2:

Remarks to the Author:

The authors have mostly addressed by queries.

Please define CSC in the abstract

Reviewer #3:

Remarks to the Author:

Having, read the comments made in response all reviewers and the revised manuscript I would no recommend acceptance of this version provided my colleague reviewers are equally satisfied that the authors have addressed their main concerns.

Dec 11, 2023

Ref: NCOMMS-22-43264A

Rebuttal to reviewers.

Reviewer #1 (Remarks to the Author):

I am happy with the revisions made and recommend acceptance.

R. We thank the reviewer for their insightful comments on our manuscript which have helped strengthen it in its revised form.

Reviewer #2 (Remarks to the Author):

The authors have mostly addressed by queries.

R. We thank the reviewer for their detailed critique of our manuscript which has helped strengthen it in its revised form.

Please define CSC in the abstract

R. The term CSC does not appear in the abstract, but it does appear in the discussion. We have replaced this term with the phrase “cancer stem cell”.

Reviewer #3 (Remarks to the Author):

Having, read the comments made in response all reviewers and the revised manuscript I would no recommend acceptance of this version provided my colleague reviewers are equally satisfied that the authors have addressed their main concerns.

R. We thank the reviewer for providing feedback which has helped strengthen our revised manuscript.

I remain concerned by the difficulty in being able to unravel the complexity of the writing and the experiments performed, even after reading the paper several times.

R. By its very nature, global transcriptomics and network analysis is complex. Indeed, both reviewers 1 and 2, and the Editor, asked that we provide further evidence of a shared regulatory network between ES cells and LSC – stressing the importance of these analyses to our study. Furthermore, we have carefully designed our experimental approaches to demonstrate that p53 has a self-renewal function in CML LSC and these are based on multiple in vivo experimental paradigms that encompass the full clinical spectrum of the disease and how an MDM2i would be administered in clinical trials. However, to address this reviewer’s concerns and improve readability, we have refined the language used to explain these analyses throughout the revised version of our manuscript. We have also addressed the reviewer’s specific concerns below, to provide further clarity on our approaches and principal findings.

Many claims are made, often based on correlative interpretations and without reference to definitive experiments.

In essence, the paper is claiming

- that quiescent LSCs are similar to ESCs: although true, I am not clear why this is important. Could it just be that various lineages share a similar quiescence program?

R. The similarity between ESC and LSC, particularly the involvement of p53 in a regulatory network shared by both cell types, is an important finding of our study. ESC utilise p53 to regulate self-renewal, and by virtue of this shared regulatory network, we hypothesized a role for p53 in self-renewal in LSC. We have been very clear in our manuscript to point out that this similarity is an important feature of our analysis and underpins the functional work which we subsequently described. We have stated this in our abstract, at the end of our introduction, in the results (first para on pg 7), and in the discussion (last para on pg 14 and first para on pg 15). We believe we have reinforced this concept throughout our manuscript.

- that MDM2i + TKI reduce CFC formation and engraftment: this is not surprising, as this combination is likely to induce proliferation arrest and cell death.

R. We acknowledge that our in vitro studies did show an increase in cell death (Fig. 3a,c; Supplementary Fig. 3a) which could explain why we observed reductions in CFC outputs obtained from these experiments. However, our in

vivo experiments clearly demonstrated that exposure to an MDM2i in combination with nilotinib did not significantly reduce stem and progenitor cell numbers at the end of treatment although CFC outputs from these experiments were significantly reduced (Fig. 4; Supplementary Fig. 4). Furthermore, reductions in engraftment potential in secondary recipients transplanted with cells exposed to this drug combination was not preceded by a significant loss of LT-HSC in the primary recipients. Thus, loss of CFC or engraftment potential cannot be accounted for by reductions in cell numbers due to cell death in vivo as postulated by the reviewer.

- that LSC exhaustion results when treatment is ceased: this should require a limiting dilution transplant experiment

R. We thank the reviewer for this feedback and note that this new experiment was not requested during the first review of our manuscript. In our serial engraftment experiment using the transgenic mouse model, we transplanted equal numbers of LT-HSC from our nilotinib only and nilotinib + MDM2i treated primary recipient mice into secondary cohorts. Therefore, the reduction in engraftment in the secondary cohort that we observed in the nilotinib + MDM2i arm when compared to the nilotinib only arm (Supplementary Fig. 5g) can only be explained by loss of LSC function (i.e., LSC exhaustion). We would also like to clarify that limiting dilution experiments using a PDX model are not possible in chronic phase CML, as successful secondary transplantation of Ph⁺ cells has never been reported in the literature.

- that MDM2i + TKI enhance capacity to enable TKI discontinuation: this may be true, but the more likely interpretation is that the combination enhances progenitor elimination and reduction of MRD.

R. We have clarified in the manuscript that the clinical value of MDM2i + TKI is not only to reduce MRD, but to functionally impair the cells that persist in MRD, thus improving the likelihood that patients will remain in remission once TKI therapy is discontinued. This clarification now appears in the Discussion (bottom of pg. 17).

I really found this paper a struggle to connect together, with many of the conclusions linked to a large amount of, often, complex data.

It is hard to know how to direct the authors to address the above without asking for a completely new design and set of experiments. As a result, I accept the authors have addressed what I have asked them to do but I remain concerned about the overall readability, interpretation and conviction of the conclusions.

R. We believe our clarifications described above and improvements to the readability of the manuscript have improved it further. We thank the reviewer for this valuable feedback.